# Induction chemotherapy plus camrelizumab followed by concurrent chemoradiotherapy in unresectable locally advanced esophageal squamous cell carcinoma: a single-arm phase II trial

Concurrent chemoradiotherapy (CCRT) has remained the standard treatment for unresectable locally advanced esophageal squamous cell carcinoma (ESCC), yet survival remains poor. This single-arm, phase II trial aims to evaluate the efficacy and safety of two cycles of induction chemotherapy with camrelizumab followed by CCRT in previously untreated patients with unresectable locally advanced ESCC. The primary endpoint, the 1-year overall survival (OS) rate in the per-protocol population (N = 46), was 87.0% (95% confidence interval [CI]: 77.7%–97.3%), exceeding the pre-specified target. Secondary endpoints included OS in the intention-to-treat (ITT) population, progression-free survival (PFS), overall response rate (ORR), disease control rate (DCR), duration of response, safety, and health-related quality of life. In the ITT population (N = 49), the 1-year OS rate was 85.7% (95% CI: 76.5%–96.1%). The 1-year PFS rates in the per-protocol and ITT populations were 71.7% (95% CI: 59.8%–86.0%) and 71.4% (95% CI: 59.8%–85.3%), respectively. The median OS, PFS, and duration of response were not reached. Following CCRT, the ORR was 93.5%, with a DCR of 95.7%. Lymphopenia was the most frequent Grade ≥3 adverse event (100%). One patient died from treatment-related myelosuppression. Health-related quality of life generally improved after induction therapy, with significant improvements in global health status, emotional functioning, and some symptom relief, despite a slight decline in physical functioning. Here, we show that induction chemoimmunotherapy followed by CCRT exhibits promising efficacy and manageable safety in patients with unresectable locally advanced ESCC, thus warranting further randomized controlled trials. Trial number: ChiCTR2000034304.

Esophageal cancer (EC) remains a significant global health challenge, with approximately 511,000 new cases and 445,000 deaths in 2022, making it the seventh leading cause of cancer-related mortality worldwide[1]. The disease is characterized by its heterogeneous nature, with esophageal squamous cell carcinoma (ESCC) being the predominant histological subtype in regions such as Asia, particularly in China, where it accounts for over 90% of all EC cases[2,3]. Due to the lack of specific early symptoms, many patients are diagnosed at advanced

✉e-mail: chengch3@mail.sysu.edu.cn; baoyong@mail.sysu.edu.cn

stage when treatment options are limited and rapid progression occurs[3,4]. For unresectable locally advanced ESCC, definitive chemoradiotherapy (CRT) remains the standard treatment. However, despite decades of use, CRT outcomes remain suboptimal, with 3-year overall survival (OS) rates ranging from 26.9% to 38.1%[5–8]. This highlights the urgent need for more effective treatment strategies to improve outcomes in this patient population.

Immunotherapy, particularly immune checkpoint inhibitors targeting the programmed cell death-1 (PD-1)/programmed cell death-ligand 1 (PD-L1) pathway, has demonstrated significant survival benefits across various cancers[9]. Anti-PD-1 monotherapy has shown promise in improving outcomes for advanced EC patients in the later-line setting[10,11]. Furthermore, chemoimmunotherapy has become a standard first-line treatment for advanced ESCC[12]. The combination of immunotherapy with CRT has also emerged as an active area of research that may provide a more effective regimen for locally advanced EC[13–15].

Various strategies exist for combining immunotherapy with radiotherapy in the treatment of EC, including pre-radiotherapy immunotherapy, concurrent radioimmunotherapy, post-radiotherapy immunotherapy, or various combinations thereof. However, the optimal timing and sequencing of immunotherapy in conjunction with CRT remain to be fully elucidated. Our previous retrospective analysis previously demonstrated that induction chemoimmunotherapy followed by definitive CRT improved survival outcomes and maintained a manageable safety profile in patients with unresectable locally advanced ESCC[16]. Nevertheless, these findings necessitate further validation through prospective studies.

Camrelizumab, a humanized, high-affinity IgG4-kappa PD-1 monoclonal antibody, has demonstrated promising antitumour activity against EC[11,17]. Based on the results from the phase 3 ESCORT-1st trial, chemotherapy plus camrelizumab has been recommended as first-line treatment for advanced ESCC[17]. Furthermore, in a phase 1b study, the addition of camrelizumab to definitive CRT for locally advanced ESCC exhibited a manageable safety profile and promising efficacy[14].

Herein, we conduct a single-arm phase 2 trial to evaluate the efficacy and safety of induction chemoimmunotherapy followed by concurrent chemoradiotherapy (CCRT) as a first-line treatment for unresectable locally advanced ESCC. In this work, the addition of chemoimmunotherapy to first-line CCRT demonstrates promising efficacy with a manageable safety profile, supporting its potential as a feasible regimen for further evaluation in phase III clinical trials.

## Results

### Patient characteristics

Between July 12, 2020, and October 14, 2022, a total of 69 patients with ESCC were screened for eligibility; 49 eligible patients with unresectable locally advanced ESCC were enrolled (Fig. 1). The baseline characteristics of the study cohort are presented in Table 1. The median age was 62 years (range: 41–74), comprising of 41 males (83.7%) and 8 females (16.3%). At baseline, stage III or IVA disease was diagnosed in 28 patients (57.1%), while 16 patients (32.7%) had stage IVB disease solely due to supraclavicular lymph node metastasis.

### Treatment compliance

All 49 patients completed the planned two cycles of induction chemotherapy combined with camrelizumab (Intention-to-Treat, ITT), with only one patient (2.0%) experiencing a reduction in chemotherapy dosage due to hepatic dysfunction. Following induction therapy, three patients did not proceed to CCRT and withdrew from the study: two refused CCRT, while one opted for surgery instead. Among the remaining 46 patients who continued with CCRT (Per-protocol set, PPS), all achieved a minimum radiation dose of 50 Gy, with 37 patients (80.4%) receiving at least 60 Gy. Only one patient (2.2%) discontinued

radiotherapy prematurely due to endotoxic shock. Premature cessation of concurrent chemotherapy occurred in two patients (4.3%) and six patients (13.0%) required dose reduction of concurrent chemotherapy. Details on treatment compliance are listed in Supplementary Table 1.

### Efficacy

As of February 1, 2024, the median follow-up duration for the 32 surviving participants was 30.7 months (range: 15.8–42.2). Among the 46 patients in the PPS, 15 deaths (32.6%) occurred. The primary endpoint of 1-year OS rate was 87.0% (95% confidence interval [CI]: 77.7–97.3%), which exceeded the pre-specified target. The 2-year OS rate was 75.8% (95% CI: 64.2–89.3%). The 1-year and 2-year progression-free survival (PFS) rates were 71.7% (95% CI: 59.8–86.0%) and 62.9% (95% CI: 50.3–78.6%), respectively. Median OS and PFS had not been reached as of the cut-off date (Fig. 2a, b). In the ITT population of 49 patients, the maturity rate for OS data was 34.7%, and the median OS was not reached (Fig. 2c). The 1-year OS rate was 85.7% (95% CI: 76.5–96.1%), and the 2-year OS rate was 72.5% (95% CI: 60.7–86.6%). The 1-year and 2-year PFS rates were 71.4% (95% CI: 59.8–85.3%) and 60.5% (95% CI: 48.1–76.2%), respectively. Median PFS was not reached (Fig. 2d).

Radiological response of 46 patients in the PPS was evaluated and presented in Table 2. After induction chemotherapy plus camrelizumab, the objective response rate (ORR) and disease control rate (DCR) were 91.3% (95% CI: 79.2–97.6%) and 100% (95% CI: 92.3–100%), respectively (Fig. 3a, b). One patient (2.2%) achieved a complete response (CR), and 41 patients (89.1%) had partial responses (PR); no patients exhibited progressive disease after induction therapy. The rate of esophageal ulcers detected via endoscopy decreased significantly from 56.5% (26/46) before treatment to 10.9% (5/46) after induction therapy; twenty-two cases of esophageal ulcers showed improvement following induction chemotherapy plus camrelizumab, while one case developed new esophageal ulcers post-treatment (p < 0.001; Supplementary Fig. 1). Furthermore, the incidence of tumor necrosis observed through enhanced computed tomography (CT) decreased from 19.6% (9/46) before treatment to 4.3% (2/46) after induction therapy; seven cases of tumor necrosis demonstrated improvement following induction therapy, while no new cases of tumor necrosis were detected post-treatment. Statistical analysis using McNemar's test revealed a significant reduction in tumor necrosis after induction (p = 0.016; Supplementary Fig. 2).

After one month of completing CCRT, the ORR reached 93.5% (95% CI: 82.1–98.6%), with a DCR of 95.7% (95% CI: 85.2–99.5%) (Fig. 3c). However, two patients died within one month after treatment completion, thereby precluding an overall assessment of efficacy. Among the 44 patients analyzed for their best response following CCRT, all exhibited a reduction in the size of the target lesions compared to baseline: seven patients (15.2%) achieved a CR, 36 (78.3%) had a PR, and one (2.2%) exhibited stable disease (SD). Importantly, treatment effects were sustained until the cutoff date in 28 of these patients (60.9%), including a significant maintenance rate of CR observed in six of seven patients (85.7%); among patients with PR or SD, it was maintained in 22 of 37 cases (59.5%; Fig. 3d). The median duration of response was not reached (95% CI: 26.6 months to not estimable).

Tumor recurrence occurred in 26.5% (13/49; 95% CI: 14.9–41.1%) of patients, including locoregional failure in 10.2% (5/49; 95% CI: 3.4–22.2%), distant metastasis in 8.2% (4/49; 95% CI: 2.3–19.6%), and simultaneous locoregional and distant failure in 8.2% (4/49; 95% CI: 2.3–19.6%). By the last follow-up, a total of 17 patients (34.7%) died, including two from the ITT population who did not receive CCRT. Recurrence patterns and causes of death are detailed in Supplementary Tables 2, 3. It is worth noting that this study coincided with the coronavirus disease 2019 (COVID-19) pandemic in China, during which two patients developed COVID-19 infection while undergoing radiotherapy. This unfortunate circumstance resulted in an interruption of

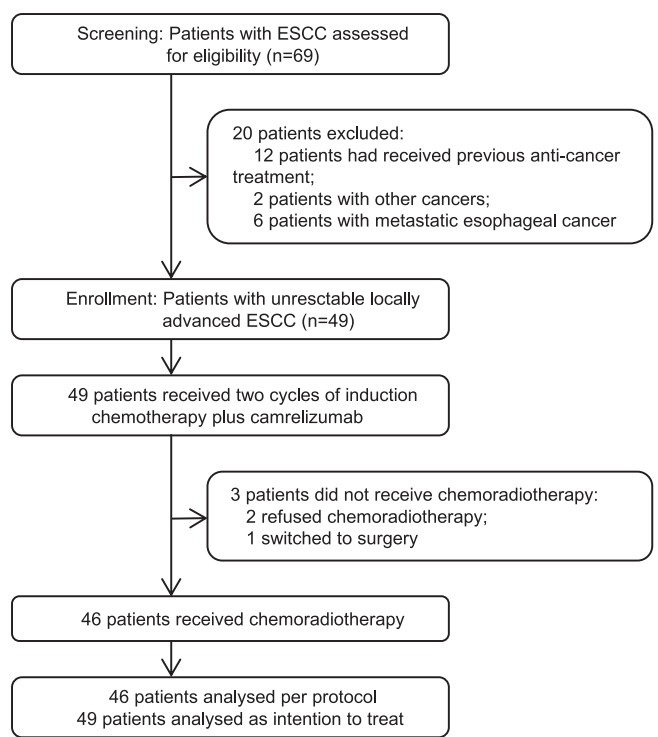

**Fig. 1 | Flow diagram of participant enrollment, treatment, and analysis.** A total of 69 patients were screened for eligibility in the trial. Twenty patients were excluded according to the inclusion and exclusion criteria. Forty-nine eligible patients were enrolled and administered two cycles of induction chemotherapy plus camrelizumab. Three patients discontinued the study treatment due to failure to receive subsequent chemoradiotherapy as per the study protocol. All 49 enrolled patients were included in the intention-to-treat analysis, while 46 patients who completed the full treatment protocol were included in the per-protocol analysis.

**Table 1 | Baseline characteristics**

| Characteristic | Patients (N = 49) |
|---|---|
| Age, years; median (Range) | 62 (41–74) |
| Sex | |
| Male | 41 (83.7%) |
| Female | 8 (16.3%) |
| Smoking history | |
| Yes | 32 (65.3%) |
| No | 17 (34.7%) |
| Alcohol history | |
| Yes | 28 (57.1%) |
| No | 21 (42.9%) |
| Bodyweight loss within 3 months | |
| <5% | 36 (73.5%) |
| ≥5% | 13 (26.5%) |
| ECOG performance status | |
| 0 | 31 (63.3%) |
| 1 | 18 (36.7%) |
| Tumor location | |
| Upper | 22 (44.9%) |
| Middle | 15 (30.6%) |
| Distal | 12 (24.5%) |
| Primary tumor length (cm) | |
| <5 | 10 (20.4%) |
| ≥5 | 39 (79.6%) |
| Clinical T stage | |
| T1-2 | 6 (12.2%) |
| T3-4 | 43 (87.8%) |
| Clinical N stage | |
| N0-1 | 17 (34.7%) |
| N2-3 | 32 (65.3%) |
| Clinical TNM stage | |
| II | 5 (10.2%) |
| III | 18 (36.7%) |
| IVA | 10 (20.4%) |
| IVB[a] | 16 (32.7%) |
| PD-L1 status | |
| CPS < 10 | 14 (28.6%) |
| CPS ≥ 10 | 19 (38.8%) |
| Unknown | 16 (32.7%) |
| Reasons for no surgery | |
| Inoperable | 25 (51.0%) |
| Cervical | 14 (28.6%) |
| Surgical contraindication | 2 (4.1%) |
| Patient refusal | 8 (16.3%) |

Data are presented as *n* (%) unless otherwise stated.
[a]Stage IVB based solely on supraclavicular lymph node metastasis. ECOG Eastern Cooperative Oncology Group, PD-L1 programmed cell death-ligand 1, CPS combined positivity score.

their radiotherapy for over two weeks, and subsequently led to significant local disease progression within one year, ultimately resulting in subsequent deaths. The sensitivity analysis demonstrated that excluding these patients did not reverse the trend and the nominal *p* value did not indicate there is a difference clinically in OS (nominal *p* = 0.914) and PFS (nominal *p* = 0.782) between the two groups, indicating the robustness of survival estimates despite their inclusion in the primary analysis (Supplementary Fig. 3). Additionally, another one patient who had been tumor-free for two years, also succumbed to a COVID-19 infection.

An exploratory post-hoc analysis using propensity-score matching (PSM) compared the trial group comprising 46 patients from the PPS in this study with the control group consisting of 71 patients who received standard definitive CRT at our center (Supplementary Table 4 and Supplementary Fig. 4). Median follow-up duration for living patients in the control cohort was 58.0 months (range: 33.3–80.1). Before PSM, the trial group exhibited significantly improved OS and PFS compared to the control group: the 2-year OS rate was 75.8% (95% CI: 64.2–89.3%) versus 50.7% (95% CI: 40.3–63.8%; HR [hazard ratio] = 0.495, 95 CI: 0.276–0.891; *p* = 0.017); the 2-year PFS rate was 62.9% (95% CI: 50.3–78.6%) versus 40.8% (95% CI: 30.9–54.0%; HR = 0.522, 95% CI: 0.304–0.898; *p* = 0.017). After PSM of 46 pairs, the 2-year OS rate was 75.8% (95% CI: 64.2–89.3%) versus 58.7% (95% CI: 46.1–74.8%; HR = 0.557, 95 CI: 0.297–1.043; *p* = 0.064); furthermore, PFS remained longer in the trial group compared to the control group (HR = 0.545, 95% CI: 0.305–0.976; *p* = 0.038), with respective 2-year PFS rate of 62.9% (95% CI: 50.3–78.6%) versus 45.7% (95% CI: 33.3–62.6%).

After the administration of induction chemotherapy combined with camrelizumab, a significant tumor shrinkage was observed in 52.2% (24/46) of patients, with those showing a reduction exceeding 50%. To investigate the impact of tumor response following induction therapy on survival outcomes, an exploratory post-hoc analysis was conducted (Supplementary Fig. 5). Based on the maximum change in tumor size from baseline following induction therapy, the 46 patients were categorized into low responders (22 cases, 47.8%) and high

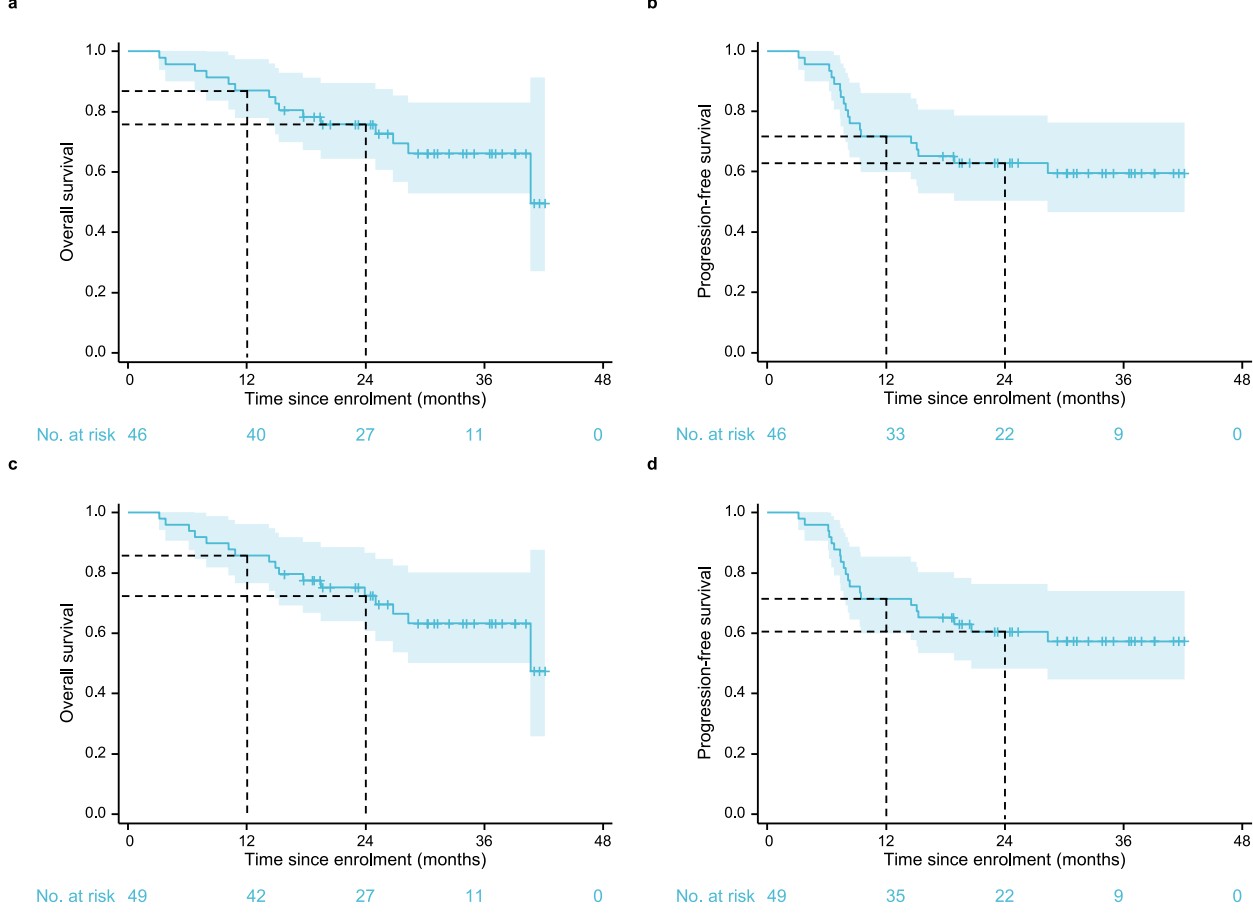

**Fig. 2 | Kaplan–Meier estimates of survival. a** Overall survival in the per-protocol set (*n* = 46). **b** Progression-free survival in the per-protocol set (*n* = 46). **c** Overall survival in the intention-to-treat population (*n* = 49). **d** Progression-free survival in the intention-to-treat population (*n* = 49). Kaplan–Meier survival curves are shown with 95% confidence intervals (shaded areas). Source data are provided as a Source Data file.

### Table 2 | Tumor response to treatment

| Treatment response | Patients (*N* = 46) |
|---|---|
| Tumor response after induction therapy | |
| Complete response | 1 (2.2%) |
| Partial response | 41 (89.1%) |
| Stable disease | 4 (8.7%) |
| Progressive disease | 0 |
| Not evaluable | 0 |
| Overall response | |
| Complete response | 7 (15.2%) |
| Partial response | 36 (78.3%) |
| Stable disease | 1 (2.2%) |
| Progressive disease | 0 |
| Not evaluable | 2 (4.3%) |

Data are presented as *n* (%). Three patients who did not receive chemoradiotherapy were excluded from the statistical analysis of tumor response. Two patients who died within one month after treatment completion were classified as 'Not evaluable' for overall response efficacy.

responders (24 cases, 52.2%) using a cut-off value of 50% as depicted in Fig. 2b. Both OS and PFS showed significantly better results in high responders compared to low responders (*p* = 0.001 for OS; *p* = 0.003 for PFS). Patients who achieved a CR after CCRT had significantly improved survival outcomes compared to those without CR (*p* = 0.046 for OS; Supplementary Fig. 6). All the seven patients who achieved a CR survived for more than 2 years, while the 2-year OS rate for 39 patients without CR was 71.3% (95% CI: 58.3–87.2%). Only one (14.3%) of seven patients with CR experienced disease progression 12.2 months after the completion of radiotherapy, whereas disease progression was observed in 17 (43.6%) of 39 patients without CR. Notably, up to 14 (82.4%) of these 17 patients developed disease progression within one year after CCRT administration.

Additionally, a post-hoc exploratory analysis was conducted to compare the survival outcomes between patients with and without supraclavicular lymph node metastasis (Supplementary Fig. 7). The results revealed no statistically significant difference in OS (HR = 0.667, 95% CI: 0.212–2.104; *p* = 0.487) or PFS (HR = 0.753, 95% CI: 0.268–2.115; *p* = 0.590).

### Safety
Treatment-related adverse events (TRAEs) are summarized in Table 3. During the induction phase, all 49 patients experienced TRAEs of any grade, with the most common ones being myelosuppression such as anemia (45 of 49, 91.8%) and lymphopenia (39 of 49, 79.6%). The most frequent grade 3 TRAEs during the induction phase included myelo-suppression such as lymphopenia (9 of 49, 18.4%) and aspartate ami-notransferase (AST) elevation (6 of 49, 12.2%). Notably, only one patient experienced a grade 4 TRAE (lymphopenia) and no grade 5 TRAEs occurred during induction therapy.

## Table 3 | Adverse events

| | Induction therapy-related (N = 49) | | | | | Treatment-related (N = 46) | | | | |
|---|---|---|---|---|---|---|---|---|---|---|
| | Grades 1–2 | Grade 3 | Grade 4 | Grade 5 | All | Grades 1–2 | Grade 3 | Grade 4 | Grade 5 | All |
| Leukopenia | 26 (53.1) | 1 (2.0) | 0 | 0 | 27 (55.1) | 26 (56.5) | 4 (8.7) | 3 (6.5) | 1 (2.2) | 34 (73.9) |
| Lymphopenia | 29 (59.2) | 9 (18.4) | 1 (2.0) | 0 | 39 (79.6) | 0 | 12 (26.1) | 34 (73.9) | 0 | 46 (100) |
| Neutropenia | 16 (32.7) | 2 (4.1) | 0 | 0 | 18 (36.7) | 21 (45.7) | 4 (8.7) | 2 (4.3) | 1 (2.2) | 28 (60.9) |
| Anemia | 41 (83.7) | 4 (8.2) | 0 | 0 | 45 (91.8) | 35 (76.1) | 11 (23.9) | 0 | 0 | 46 (100) |
| Thrombocytopenia | 10 (20.4) | 0 | 0 | 0 | 10 (20.4) | 15 (32.6) | 6 (13.0) | 5 (10.9) | 0 | 26 (56.5) |
| ALT elevation | 24 (49.0) | 4 (8.2) | 0 | 0 | 28 (57.1) | 24 (52.2) | 6 (13.0) | 0 | 0 | 30 (65.2) |
| AST elevation | 20 (40.8) | 6 (12.2) | 0 | 0 | 26 (53.1) | 21 (45.7) | 6 (13.0) | 0 | 0 | 27 (58.7) |
| GGT elevation | 27 (55.1) | 2 (4.1) | 0 | 0 | 29 (59.2) | 31 (67.4) | 2 (4.3) | 0 | 0 | 33 (71.7) |
| ALP elevation | 18 (36.7) | 0 | 0 | 0 | 18 (36.7) | 36 (78.3) | 0 | 0 | 0 | 36 (78.3) |
| Hyperbilirubinemia | 10 (20.4) | 0 | 0 | 0 | 10 (20.4) | 9 (19.6) | 0 | 0 | 0 | 9 (19.6) |
| Creatinine increased | 2 (4.1) | 0 | 0 | 0 | 2 (4.1) | 6 (13.0) | 0 | 0 | 0 | 6 (13.0) |
| Hypothyroidism | 0 | 0 | 0 | 0 | 0 | 0 | 0 | 0 | 0 | 0 |
| Pruritus | 20 (40.8) | 0 | 0 | 0 | 20 (40.8) | 20 (43.5) | 0 | 0 | 0 | 20 (43.5) |
| Peripheral sensory neuropathy | 25 (51.0) | 1 (2.0) | 0 | 0 | 26 (53.1) | 24 (52.2) | 1 (2.2) | 0 | 0 | 25 (54.3) |
| Fever | 2 (4.1) | 0 | 0 | 0 | 2 (4.1) | 12 (26.1) | 0 | 0 | 0 | 12 (26.1) |
| Rash | 4 (8.2) | 0 | 0 | 0 | 4 (8.2) | 4 (8.7) | 0 | 0 | 0 | 4 (8.7) |
| RCCEP | 12 (24.5) | 0 | 0 | 0 | 12 (24.5) | 12 (26.1) | 0 | 0 | 0 | 12 (26.1) |
| Nausea | 11 (22.4) | 0 | 0 | 0 | 11 (22.4) | 28 (60.9) | 1 (2.2) | 0 | 0 | 29 (63.0) |
| Vomiting | 4 (8.2) | 0 | 0 | 0 | 4 (8.2) | 17 (37.0) | 1 (2.2) | 0 | 0 | 18 (39.1) |
| Diarrhoea | 7 (14.3) | 1 (2.0) | 0 | 0 | 8 (16.3) | 14 (30.4) | 0 | 0 | 0 | 14 (30.4) |
| Oral mucositis | 0 | 0 | 0 | 0 | 0 | 16 (34.8) | 0 | 0 | 0 | 16 (34.8) |
| Constipation | 20 (40.8) | 0 | 0 | 0 | 20 (40.8) | 42 (91.3) | 0 | 0 | 0 | 42 (91.3) |
| Weight loss | 6 (12.2) | 0 | 0 | 0 | 6 (12.2) | 25 (54.3) | 0 | 0 | 0 | 25 (54.3) |
| Alopecia | 18 (36.7) | 0 | 0 | 0 | 18 (36.7) | 18 (39.1) | 0 | 0 | 0 | 18 (39.1) |
| Pneumonitis | 4 (8.2) | 0 | 0 | 0 | 4 (8.2) | 16 (34.8) | 0 | 0 | 0 | 16 (34.8) |
| Esophageal fistula | 0 | 0 | 0 | 0 | 0 | 0 | 0 | 0 | 0 | 0 |
| Radiation dermatitis | NA | NA | NA | NA | NA | 14 (30.4) | 0 | 0 | 0 | 14 (30.4) |
| Radiation esophagitis | NA | NA | NA | NA | NA | 42 (91.3) | 2 (4.3) | 0 | 0 | 44 (95.7) |
| Cough | NA | NA | NA | NA | NA | 36 (78.3) | 0 | 0 | 0 | 36 (78.3) |

Data are presented as in (%).
*ALT* alanine aminotransferase, *AST* aspartate aminotransferase, *GGT* gamma-glutamyl transferase, *ALP* alkaline phosphatase, *RCCEP* reactive cutaneous capillary endothelial proliferation, *NA* not applicable.

Among the 49 patients, the most common immune-related adverse events (AEs) observed were pruritus in 20 patients (40.8%), reactive cutaneous capillary endothelial proliferation (RCCEP) in 12 patients (24.5%), and rash in four patients (8.2%), all classified as grade 1–2. These aforementioned AEs occurred during induction therapy, and all patients recovered without severe complications.

Throughout the treatment period for all 46 evaluable patients in the PPS, lymphopenia was the most common grade 3 or higher TRAE in all 46 patients (100%). Additionally, anemia (11 of 46, 23.9%), thrombocytopenia (11 of 46, 23.9%), leukopenia (8 of 46, 17.4%), neutropenia (7 of 46, 15.2%), alanine aminotransferase (ALT) elevation (6 of 46, 13.0%), and AST elevation (6 of 46, 13.0%) were also common among these patients. The incidence rate of grade 3 or higher radiation esophagitis was 4.3% (2 of 46). Notably, no cases of esophageal fistula occurred during the study period. Furthermore, there were no instances recorded of grade 3 or higher pneumonitis. One (2.2%) patient died of infection due to myelosuppression within one month after CCRT. Another patient died of systemic multiple organ failure subsequent to self-administering Chinese medicine with unknown composition after finishing treatment; however, this event was not deemed directly related to the treatment.

### Health-related quality of life

Health-related quality of life was assessed and compared between baseline and post-induction therapy using the European Organization for Research and Treatment of Cancer Quality of Life Questionnaire core 30 (QLQ-C30) and the Quality of Life Questionnaire-Esophageal Cancer Module 18 (QLQ-OES18). Completion rate for QLQ-C30 and QLQ-OES18 was 71.4% (35/49) and 79.6% (39/49), respectively. Compared to baseline, global health status showed significant improvement after induction therapy ($p = 0.007$). Similarly, there was a significant increase in emotional functioning from baseline to post-induction therapy ($p = 0.003$). However, patients' physical functioning slightly worsened after induction therapy ($p = 0.043$). In terms of symptoms, pain ($p = 0.012$ in QLQ-C30; $p < 0.001$ in QLQ-OES18), choking when swallowing ($p = 0.005$), trouble with coughing ($p = 0.005$), and trouble with eating ($p = 0.048$) were significantly alleviated after induction therapy. No statistical differences were observed in other metrics of QLQ-C30 and QLQ-OES18 compared to baseline (Supplementary Fig. 8).

### Tumor PD-L1 expression

An exploratory analysis was performed exploring the association between PD-L1 expression status and clinical outcomes. PD-L1

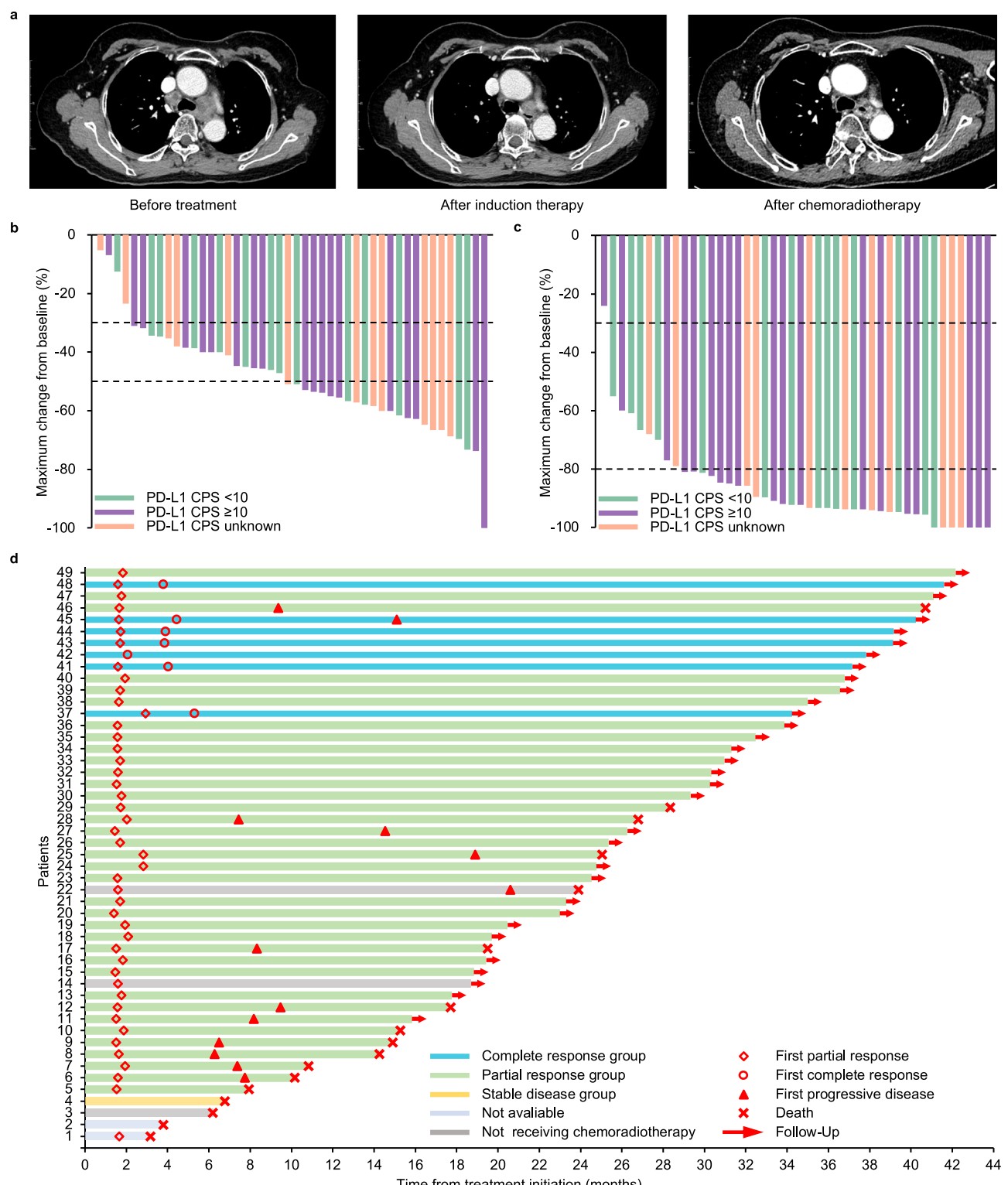

**Fig. 3 | Tumor responses. a** Contrast-enhanced CT images from a representative patient showing significant tumor shrinkage following induction chemotherapy plus camrelizumab and sustained partial response after concurrent chemoradiotherapy. **b** Maximum change in target lesion sizes from baseline assessed per RECIST 1.1 after induction chemotherapy plus camrelizumab ($n = 46$). Three patients who did not receive chemoradiotherapy were excluded from the 49 intention-to-treat patients. **c** Maximum change in target lesion sizes from baseline assessed per RECIST 1.1 after concurrent chemoradiotherapy ($n = 44$). Two patients who died before overall response evaluation were excluded from the 46 per-protocol patients. **d** Onset of response, duration of response, and outcome ($n = 49$). Tumor response was assessed per RECIST 1.1. Source data are provided as a Source Data file. CT computed tomography, PD-L1 programmed cell death-ligand 1, CPS combined positive score, RECIST Response Evaluation Criteria in Solid Tumors.

expression was evaluable in 33 patients for overall response, with 19 (57.6%) patients having a combined positive score (CPS) of 10 or greater. CR rate was 15.8% (3/19) in patients with PD-L1 CPS ≥ 10 and 7.1% (1/14) in those with PD-L1 CPS < 10, with no statistically significant difference ($p = 0.620$). Regarding maximum change in tumor size from baseline, a decline of over 80% was observed in 84.2% (16/19) of patients with PD-L1 CPS ≥ 10 and in 71.4% (10/14) of patients with PD-L1 CPS < 10, with no statistically significant difference ($p = 0.422$; Fig. 2c). Similarly, there was no significant difference in OS ($p = 0.487$) or PFS ($p = 0.989$; Supplementary Fig. 9).

### Immune microenvironment

The tumor microenvironment was assessed in 35 patients using multiplex immunofluorescence staining for cluster of differentiation 4 (CD4), cluster of differentiation 8 (CD8), pan cytokeratin (pan-CK), C-C chemokine receptor 7 (CCR7), and death receptor 5 (DR5), revealing a significant increase in the infiltration of CD8+ and DR5+ cells (Supplementary Figs. 10, 11). With the exception of DR5 density, none of the immune cell populations in the tumor microenvironment showed an association with tumor response following induction chemotherapy plus camrelizumab (Supplementary Fig. 12). An exploratory analysis was conducted to investigate the association between the expression of various immune cell infiltration and treatment outcomes (Supplementary Fig. 13). There was a strong correlation of high CD8 density with improved OS ($p = 0.014$) and PFS ($p = 0.023$; Supplementary Fig. 13c, d). Importantly, there was a higher abundance of infiltrating DR5+ cells observed in the high responders compared to the low responders ($p = 0.02$); Additionally, patients with a high density of DR5 showed a significant improvement in OS ($p = 0.017$) and PFS ($p = 0.028$) compared to those with a low density of DR5 (Supplementary Fig. 13i, j).

### Circulating lymphocyte subsets and cytokines

Circulating lymphocyte subsets and cytokines were assessed to investigate their impact on survival outcomes during treatment. Flow cytometry was utilized to measure various lymphocyte subsets, including total T lymphocytes (CD3+CD19−), helper T cells (CD3+CD4+), cytotoxic T lymphocytes (CD3+CD8+), B lymphocytes (CD3-CD19+), and natural killer (NK) cells (CD3-CD16+CD56+). At baseline, no association between any circulating lymphocyte subset or cytokine and tumor response following induction chemotherapy plus camrelizumab was observed (Supplementary Figs. 14, 15). The count of CD3+CD8+T lymphocytes at baseline did not significantly affect PFS ($p = 0.119$); however, a significant correlation was observed between high counts of CD3 + CD8+T lymphocytes and improved OS ($p = 0.041$; Supplementary Fig. 16e, f). Although variations were observed in the levels of CD3+CD19- T lymphocytes, CD3+CD4+ helper T cells, CD3−CD19+B lymphocytes, and NK cells, these differences did not have a significant impact on survival outcomes (Supplementary Fig. 16). Similarly, the detection of cytokines such as interleukin (IL)−2, IL-4, IL-6, IL-10, tumor necrosis factor-alpha (TNF-α), and interferon-gamma (IFN-γ) also did not influence significantly survival outcomes (Supplementary Fig. 17).

## Discussion

Immunochemotherapy has significantly altered the treatment landscape for patients with advanced ESCC[12,17–20]. However, its integration with CRT in the management of unresectable locally advanced ESCC has been less explored. Our findings suggest that induction chemotherapy plus camrelizumab followed by CCRT yields promising survival outcomes and a manageable safety profile in this high-risk population, which maybe an alternative treatment strategy for patients with unresectable locally advanced ESCC. This study comprehensively elucidates the potential clinical benefits of this induction chemoimmunotherapy, including substantial tumor shrinkage, improved

tumor necrosis, and improved feeding capability, which are conducive to subsequent CCRT. Additionally, the significant healing of tumor ulcers after induction chemoimmunotherapy significantly reduces the incidence of esophageal fistula following radiotherapy, thereby minimizing post-radiotherapy adverse reactions. These findings clearly underscore the value of this induction therapy and contribute to facilitating the implementation of CCRT. These results may have implications for revised treatment protocols, incorporating earlier administration of chemoimmunotherapy as part of the therapeutic regimen for patients with unresectable locally advanced ESCC.

The primary endpoint of 1-year OS was successfully met, with an OS rate of 87.0%, surpassing the pre-specified threshold of 80.0%. Despite encountering the COVID-19 pandemic in China during this study, which adversely affected the survival of certain patients, these outcomes surpassed those reported in previous studies focusing solely on definitive CRT, which reported 1-year OS rate ranging from 52% to 66%[5,21,22]. Importantly, our trial cohort exhibited better survival outcomes when compared to patients receiving standard CRT alone from the external control cohort at our center. The trial group demonstrated significantly higher 2-year OS (75.8% vs. 50.7%) and PFS (62.9% vs. 40.8%) rates than the control group. These findings suggest that induction chemotherapy combined with camrelizumab, followed by CRT, may provide a more effective alternative to definitive CRT alone in patients with unresectable locally advanced ESCC. Notably, at the time of this analysis, the median OS and PFS are not reached yet, indicating the potential durable response in a substantial proportion of patients.

For patients with inoperable locally advanced ESCC, the role of induction chemotherapy in improving survival outcomes remains a subject of debate[8,23–25]. According to the findings from a randomized phase 2 trial, there was no significant improvement observed in response rate or survival among patients who received induction chemotherapy; however, those who responded well to induction chemotherapy demonstrated significantly better survival outcomes[8]. Immunotherapy has emerged as an integral part of EC treatment and was expected to further enhance response rate. In patients with advanced ESCC undergoing chemoimmunotherapy, the ORR has reached as high as 70%[17,19,20,26]. Moreover, operable ESCC patients receiving neoadjuvant chemoimmunotherapy exhibited a significant pathological complete response rate of 15.4% to 57.1%[27–29]. In our study, we achieved an ORR of 91.3% after induction chemotherapy plus camrelizumab, significantly superior to studies evaluating the combination of chemoimmunotherapy in advanced ESCC patients. Furthermore, our exploratory analysis revealed that patients with a high tumor response after induction therapy experienced significantly better OS and PFS outcomes compared to low responders, emphasizing the importance of achieving a robust early response.

The current study's incorporation of induction chemoimmunotherapy is notable for its dual mechanism of action, targeting both tumor cells and the tumor microenvironment. Our findings demonstrated that induction chemoimmunotherapy significantly reduced tumor volume, improved tumor necrosis, and enhanced patients' quality of life. These outcomes are conducive to increasing the efficacy of subsequent CCRT, aligning with recent literature highlighting the potential synergistic effect between immunotherapy and conventional CRT against ESCC[30]. The improved response rate observed may be attributed to immune system activation, vascular normalization, and amelioration of hypoxia, resulting in a more robust anti-tumor effect when concurrent treatment is administered.

The combination of immunotherapy with CRT has emerged as a pivotal area of research in the therapeutic protocols for inoperable locally advanced EC. Nonetheless, various combination modalities exist. Findings from small-scale studies have shown the promising efficacy of CCRT in combination with concurrent or/and adjuvant immunotherapy[13–15]. In a single-arm phase 2 study evaluating the

efficacy and safety of concurrent and consolidative durvalumab plus tremelimumab in addition to CCRT, an impressive 2-year OS rate of 75% was observed[13]. In another phase 1b trial evaluating camrelizumab combined with CCRT, the 1-year OS and PFS rates were 85.0% and 80.0%, respectively[14]. Similarly, the EC-CRT-001 trial reported a 1-year OS rate of 78.4% and a PFS 1-year rate of 54.5%[15]. However, definitive confirmation from phase 3 trials is still pending; several trials such as ESCORT-CRT, KEYNOTE-975, KUNLUN, RATIONALE-311, KYSCRAPER-07 are ongoing, but none have yet reported mature survival outcomes to validate these combination strategies[31–34]. Among them, KEYNOTE-975 is a pivotal global, randomized, double-blind phase III trial investigating pembrolizumab versus placebo in combination with definitive CRT for locally advanced unresectable EC[32]. In contrast to the concurrent administration of immunotherapy and CRT in KEYNOTE-975, our study uniquely explored induction chemoimmunotherapy followed by CRT. This sequential approach may prime the tumor microenvironment through early immune activation, potentially enhancing CRT efficacy while mitigating toxicity. Notably, the definitive CRT regimen in KEYNOTE-975 comprised one of three options for choosing, one of which delivers a total radiation dose of 60 Gy (FP including cisplatin + 5-fluorouracil with radiotherapy at 60 Gy), underscoring the clinical rationale for individualized dose escalation in specific scenarios, such as bulky tumors or poor response to induction therapy, as implemented in our trial. These collective findings highlight the need for further exploration of optimal sequencing and dosing strategies in immunotherapy-CRT combinations. It is noteworthy that our study included 32.7% of patients with M1 disease confined to supraclavicular lymph node metastases, who were not considered in the previous studies investigating concurrent and consolidative immunotherapy alongside CCRT. Our findings demonstrate promising OS and PFS rates along with an impressive ORR of 93.5% following CCRT, which highlights the therapeutic potential of combining induction chemoimmunotherapy and CCRT for locally advanced ESCC.

In terms of toxicity, the most common grade 3 or above TRAEs observed were myelosuppression, consistent with that of prior studies evaluating CRT and immunotherapy. Notably, our incidence of grade 3 or worse esophagitis was relatively low (4.3%), compared with several previous reports on definitive CRT alone, such as JCOG0909 (19.1%) and JCOG0303 (23%)[22,35]. This favorable outcome could potentially be attributed to the significant tumor shrinkage achieved during the induction phase. Importantly, despite a high rate of tumor ulceration before treatment (56.5%), none of the patients developed esophageal fistula in our study, which significantly contrasts with the historical rate after CRT[36], which may also be due to the obvious healing of tumor ulcer after induction chemoimmunotherapy. These results underscore both improved efficacy and reduced risk of severe toxicities commonly associated with conventional CRT when incorporating chemoimmunotherapy into treatment regimens. Furthermore, no grade 3 or worse radiation pneumonitis was observed in our study, indicating that the addition of induction chemotherapy plus camrelizumab did not increase the risk of severe pulmonary toxicity. Regarding immune-related toxicity, there was no grade 3 or above toxicity. Overall, the toxicity of our treatment strategy was generally manageable.

The role of biomarkers in the management of ESCC is increasingly relevant, especially in the context of immunotherapy. Our findings indicated that baseline PD-L1 expression did not significantly impact response rate or survival outcomes. These results align with a previous study[15]. Further investigation is warranted to explore the utility of PD-L1 status as a biomarker for predicting response to immunotherapy in locally advanced ESCC. Additionally, our study revealed that a high baseline density of CD8+ cells within the tumor microenvironment was indicative of treatment activity, consistent with findings reported by Zhu et al[15]. Moreover, we observed a significant correlation between elevated counts of circulating CD3+CD8+ T lymphocytes and improved OS. Therefore, measuring circulating levels of pre-treatment CD3+CD8+ cells may serve as prognostic and predictive biomarkers. Furthermore, TNF-related apoptosis-inducing ligand (TRAIL) plays an important role in apoptosis and tumor immunosurveillance. TRAIL is mainly secreted by tumor cells and binds to its specific receptor DR5 in T cells, thereby inhibiting the activation of T cells. In our previous research, we found that TRAIL expression in tumor cells was negatively correlated with the response of ESCC to neoadjuvant immunotherapy and chemotherapy[37]. DR5 serves as a receptor for immune cytokine TRAIL and plays a crucial role in the extrinsic apoptotic pathway[38,39]. Interestingly, in our current study, we found that patients with high DR5 density demonstrated substantial improvements in both OS and PFS, suggesting potential predictive value of DR5 in combination therapy involving immunotherapy and CRT. A comprehensive understanding of these biomarkers could provide valuable insights into patient stratification strategies enabling personalized therapeutic approaches that harness the full potential of immunotherapy.

While our study presents promising results, several limitations warrant consideration. First, as a single-arm phase 2 trial, the absence of a randomized control group limits the strength of the conclusions. Although we used historical controls to analyze differences in survival, a randomized controlled trial would provide a more robust framework for evaluation. Second, the absence of positron emission tomography-computed tomography (PET-CT) for overall response assessment somewhat limited our ability to confirm CR. Third, the follow-up duration remained relatively short, with median PFS and OS not yet reached. Additionally, the wide confidence intervals observed in the Kaplan-Meier survival curves highlight the limitations due to the relatively small sample size, warranting validation in larger, multicenter trials. Finally, as there is no formal hypothesis set for exploratory biomarker analyses, the *p* values are all nominal, showing a trend for further demonstration.

In conclusion, the combination of induction chemotherapy plus camrelizumab followed by CCRT demonstrated promising efficacy and manageable toxicity in patients with unresectable locally advanced ESCC. The observed improvement in survival outcomes compared to historical controls suggests potential benefits of this integrated therapeutic approach. Future randomized phase 3 trials are warranted to validate this regimen as a first-line treatment for patients with unresectable locally advanced ESCC.

## Methods
### Study design and participants
The ImpactCRT trial was a prospective, single-arm, phase 2 study conducted at the First Affiliated Hospital of Sun Yat-sen University. The study protocol is presented as Supplementary Note in the Supplementary information file. Eligible patients met the following criteria: (1) aged between 18–75 years; (2) histologically confirmed ESCC; (3) no prior treatment; (4) stage cT1–4bN0–3M0 (not suitable for surgery, including inoperable, surgical contraindication, or refusal of surgery) or M1 disease confined to supraclavicular lymph node metastases according to the 8th TNM staging system of the American Joint Committee on Cancer (AJCC); (5) presence of at least one evaluable lesion according to Response Evaluation Criteria In Solid Tumors (RECIST), version 1.1; (6) Eastern Cooperative Oncology Group (ECOG) performance status of 0–1; and (7) estimated life expectancy of at least 12 weeks. Patients also required adequate hematologic, cardiac, pulmonary, hepatic, and renal function. Key exclusion criteria included: history of other malignancies, esophageal perforation, significant tumor bleeding, serious comorbidities, severe active infection, congenital or acquired immunodeficiency, or psychiatric disorder. Sex was not considered in the study design, and the sex of participants was determined based on self-report. Both male and female participants who met the inclusion/exclusion criteria were eligible for this study.

The study was conducted in accordance with the Declaration of Helsinki and Good Clinical Practice guidelines and was approved by the Ethics Committee of the Guangdong Association Study of Thoracic Oncology. All participants provided written informed consent prior to enrollment. The trial is registered at the Chinese Clinical Trial Registry (ChiCTR.org.cn) under the identifier ChiCTR2000034304.

## Procedures

During the induction phase, patients received camrelizumab (200 mg), albumin-bound paclitaxel (260 mg/m²) and carboplatin (area under the curve [AUC] 5 mg/mL/min) on day 1 of each 21-day cycle for 2 cycles.

CCRT was administered 3–4 weeks after completing induction therapy. This included two cycles of cisplatin (75 mg/m² bolus on day 1) and fluorouracil (750 mg/m²/24 h for 5 days) during thoracic radiotherapy, repeated every four weeks. Radiotherapy was delivered using simultaneous integrated boost intensity-modulated radiotherapy (SIB-IMRT) with a linear accelerator (≥6 MV). The gross tumor volume (GTV) encompassed primary esophageal tumor (GTVp) and positive lymph nodes (GTVn). The clinical target volume (CTV) encompassed GTVp along with a superior and inferior expansion of 3 cm along the length of the esophagus, as well as a radial expansion of 0.5–1.0 cm, and GTVn plus a margin of 0.5–1.0 cm including coverage of elective nodal regions. Elective treatment of node-bearing regions depends on the location of the primary tumor in the esophagus and esophagogastric junction cancers (EGJ). Cervical esophagus: Consider treatment of the supraclavicular nodes and treatment of higher echelon cervical nodes, especially if the nodal stage was N1 or greater. Proximal third of the esophagus: Consider treatment of para-esophageal lymph nodes and supraclavicular lymph nodes. Middle third of the esophagus: Consider treatment of para-esophageal lymph nodes. Distal third of esophagus and EGJ: Consider para-esophageal, lesser curvature, and celiac axis nodal regions. Both GTV and CTV were expanded by 0.6–0.8 cm to generate the planning gross tumor volume (PGTV) and planning target volume (PTV). The prescribed dose was 50–50.4 Gy in 25–28 fractions (5 days per week) daily to PTV, and 50–63 Gy in 25–28 fractions to PGTV. Dose constraints for organs at risk were as follows: mean lung dose <17 Gy, lung V30 (percentage of the total lung volume receiving ≥30 Gy) < 20%, lung V20 (percentage of the total lung volume receiving ≥20 Gy) < 30%, lung V5 (percentage of the total lung volume receiving ≥5 Gy) < 65%, mean heart dose ≤26 Gy, heart V30 (percentage of the total heart volume receiving ≥30 Gy) < 40%, and maximum spinal cord dose ≤45 Gy. The details of quality assurance for radiotherapy are presented in the trial protocol appendix.

Pre-treatment evaluations included medical history, physical examinations, laboratory tests, barium esophagogram, esophagogastroscopy with endoscopic ultrasound, contrast-enhanced CT scan, PET-CT scan, esophageal magnetic resonance imaging (MRI) scan, pulmonary function test, electrocardiogram, and echocardiography. PET-CT was optional rather than mandatory. Bronchoscopy was performed when the tumor was suspected of invading the trachea or bronchus on CT or endoscopic ultrasound. Tumor response was evaluated prior to the onset of radiotherapy and one month after completing CCRT, according to RECIST guideline (version 1.1)[40]. Tumor response to induction therapy was assessed primarily by neck, chest, and upper abdomen enhanced CT, esophageal MRI, and esophagogastroscopy with endoscopic ultrasound, while tumor response after CRT was assessed primarily by CT and MRI. The response of the primary esophageal tumor was based on the vertical length and maximal transverse thickness of the tumor, as defined by Conroy et al[6]. Follow-up evaluations were conducted every 3 months until disease progression. AEs were systematically assessed and graded weekly according to the National Cancer Institute Common Terminology Criteria for Adverse Events, version 5.0.

## Quality of life

Health-related quality of life was assessed between baseline and post-induction therapy using QLQ-C30 and QLQ-OES18. The EORTC QLQ-C30 scale consists of 30 items that are combined to form five functioning scales (physical, role, cognitive, emotional, and social), three symptom scales (fatigue, pain, and nausea or vomiting), a global health status quality-of-life scale, as well as six single-item scales (dyspnea, insomnia, appetite loss, constipation, diarrhea, and financial difficulties). The EORTC QLQ-OES18 scale contains 18 items for patients with esophageal cancer, forming 10 symptom scales: pain, reflux, dysphagia, dry mouth, choked when swallowing, and trouble with coughing, eating, swallowing saliva, tasting, and talking.

## Immunohistochemistry

The archived baseline tumor tissue samples were analyzed for PD-L1 expression using immunohistochemistry. A commercially available PD-L1 immunohistochemistry assay (clone 22C3; DAKO Autostainer Link48; RTU) was utilized to evaluate the PD-L1 combined positive score in formalin-fixed tumor diagnostic samples, following both the manufacturer's instructions and international guidelines. Samples were considered PD-L1-positive if CPS ≥ 10 of tumor cells showed membranous PD-L1 expression. In cases where multiple pre-treatment specimens were available for PD-L1 testing, a patient was considered PD-L1-positive if any of the pre-treatment specimens were positive, and the highest percentage of PD-L1 positive tumor cells is reported here.

## Multiplex immunofluorescence staining and multispectral imaging analysis

The archived baseline tumor tissue samples were analyzed for tumor microenvironment using multiplex immunofluorescence staining. The 4μm-thick slides cut from the Formalin-fixed paraffin embedded (FFPE) blocks were dewaxed in xylene, rehydrated through a decreasing ethanol series and fixed in NBF (10% neutral buffered formalin) for 10 min. Slides were stained to enable the simultaneous visualization of five markers: Abs anti-CD8 (Cat# ab237709, Abcam), anti-CD4 (Cat# ab133616, Abcam), anti-CCR7 (Cat#ab253187, Abcam), anti-DR5 (Cat# ab8416, Abcam), anti-pan-CK (Cat# Kit-0009, MXB Biotechnologies). At the beginning of each staining cycle, microwave-heated treatment in EDTA solution was applied to perform antigen retrieval. After blocking proteins for 10 min, these five primary antibodies were sequentially incubated for 30, 30, 30, 60, 60 min at 37 °C, respectively. Then, the incubation of HRP-conjugated secondary antibody and tyramide signal amplification (TSA) with Opal was followed. Five staining cycles were performed for the following antibodies/fluorescent dyes combinations: anti-CD8/Opal-690, anti-CD4/Opal-540, anti-CCR7/Opal-620, anti-DR5/Opal-520, anti-pan-CK/Opal-570. Microwave treatment was performed at each cycle of staining to remove the Ab TSA complex. Finally, all slides were stained with 4′–6′-diamidino-2-phenylindole (DAPI, SIGMA-ALDRICH) for 8 min and enclosed with Mounting Medium 0022001010 (Panovue, Beijing, China). The stained slides were scanned using the TissueFAXS platform (TissueGnostics, Vienna, Austria) at a magnification of 20×. This platform captures fluorescent spectra at 20-nm wavelength intervals ranging from 420 to 720 nm with consistent exposure time. The resulting scans were combined to build a single stack image. Spectral libraries were created from the extracted images, incorporating both unstained and single-stained slides, which were applied to extract tissue autofluorescence spectra and individual fluorescein spectra, respectively. These libraries were then utilized for unmixing multispectral images (six-color staining) employing StrataQuest software (TissueGnostics, Vienna, Austria). By utilizing this spectral library, reconstructed images devoid of autofluorescence were obtained for subsequent imaging analysis. For each primary antibody used, the positivity cut-off value was determined based on the staining pattern and intensities observed across all images.

## Flow cytometry

Fresh blood samples were expeditiously transported to the laboratory for analysis; flow cytometry was employed to detect circulating cytokines and lymphocyte subsets according to the manufacturer's instructions. The patient blood samples were collected in 10 ml EDTA tubes and immediately centrifuged at $200 \times g$ for 30 min at 4 °C without brake. The BD FACSCanto II flow cytometer (Becton Dickinson) was employed to analyze lymphocyte subpopulations and T helper 1 (Th1)/Th2 cytokines in the venous blood anticoagulated with EDTA, following the manufacturer's instructions. The lymphocyte subpopulation analysis was performed using a BD Multitest 6-color TBNK kit (Becton Dickinson), which included total T lymphocytes (CD3+CD19- T lymphocytes), helper T lymphocytes (CD3+CD4+ T lymphocytes), cytotoxic T lymphocytes (CD3+CD8+ T lymphocytes), B lymphocytes (CD3-CD19 + B lymphocytes), and natural killer (NK) cells (CD3-CD16+CD56+ NK cells). The cytokine analysis was conducted using a Human Th1/Th2 subgroup detection kit from Hangzhou Saikey Biotechnology Co., LTD., which included interleukin (IL)−2, IL-4, IL-6, IL-10, tumor necrosis factor (TNF)-α, and interferon-gamma (IFN-γ).

## Outcomes

The primary endpoint was 1-year OS rate in PPS. Secondary endpoints included 1-year OS rate in ITT population, OS, PFS, ORR, DCR, duration of response, safety, and health-related quality of life. Additionally, an exploratory analysis was conducted to investigate the potential association of tumor tissue and/or blood biomarkers with treatment efficacy.

## Statistical analysis

A total of 44 patients were needed to observe an improvement of 14% in 1-year OS rate (from 66% in the previous study[21] to 80% in the current study), with a one-sided α level of 0.05, power of 80%, an accrual period of 18 months, and a minimum follow-up period of 12 months. Assuming a dropout rate of 10%, the final estimated sample size was determined as 49 patients.

Efficacy was analyzed in both the ITT population (all patients with at least one dose of treatment) and PPS (patients who received induction therapy and CCRT per protocol). The safety analysis during induction therapy included all patients in the ITT population, while the safety analysis throughout the entire treatment course was performed in the PPS. Continuous variables were compared between groups using the Mann-Whitney U test, while categorical variables were analyzed using Pearson's chi-squared test or Fisher's exact test. Within the same group, changes in continuous and categorical variables before and after treatment were assessed using the Wilcoxon signed rank test and the McNemar test, respectively. The 95% confidence intervals (CIs) for proportions, based on the binomial distribution, were calculated using the Clopper-Pearson method. The Kaplan-Meier method was employed to estimate OS and PFS along with their corresponding 95% CIs. A log-rank test was performed to assess survival differences between groups, and Cox proportional hazards regression was used to estimate hazard ratios (HRs) and 95% CIs. Two-sided $p < 0.05$ was considered statistically significant. R software (version 4.2.1), SPSS Statistics (version 22.0), Microsoft Excel, and TissueFAXS platform (TissueGnostics, Vienna, Austria) were used during data collection, data analysis, and figure generation.

## Reporting summary

Further information on research design is available in the Nature Portfolio Reporting Summary linked to this article.

## Data availability

Detailed individual data are available under restricted access for both legal and ethical concerns. Requests for access to de-identified participant data from this study can be submitted via email to baoyong@mail.sysu.edu.cn, accompanied by a detailed proposal for approval. Please allow 1 month for a response to the request. All data requests will undergo review by the First Affiliated Hospital of Sun Yat-Sen University to assess any potential intellectual property or confidentiality obligations. A proposal detailing the study objectives and statistical analysis plan will be required for evaluation. Data will be available upon request 12 months after the publication of this article. The raw identifying individual participant data are protected and are not available due to data privacy laws. The study protocol is available as Supplementary Note in the Supplementary Information file. The remaining data are available within the Article, Supplementary Information, or Source Data file. Source data are provided with this paper.

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

## Acknowledgements

The authors are grateful to all patients and their families. The research was financially supported by Beijing Life Oasis Public Service Center Research Foundation (grant number: H2023027) and Hengrui Pharmaceutical. However, the funders had no role in study design, data collection, data analysis, data interpretation, or writing of the report. Additionally, we would like to appreciate ChatGPT for their assistance in English language editing.

## Author contributions

Y.B. and C.C. were responsible for the conception and design of the study. Y.B. was responsible for funding acquisition. F.P., J.W., Y.B., S.W., S.N., X.X., W.S., Y.L., H.Y., S.F., X.W., W.C., and W. Ye enrolled and treated patients. F.P., J.W., H.L., W. Yang, T.Y., and W.L. contributed to data collection. F.P. and J.W. contributed to statistical analysis. All authors were responsible for the interpretation of data and writing the manuscript, as well as reviewing and approving the manuscript for submission.

## Competing interests

The authors declare no competing interests.

## Additional information

Fang Peng [1,10], Jialiang Wu [2,10], Huimin Lian[1], Shuang Wu [1], Shaoqing Niu [1], Xiangbin Xing [3], Weixiong Yang [4], Wu Song[5], Yin Li[5], Honglan Yu[5], Shi-Ting Feng [6], Xiaoyan Wang [7], Wenfang Chen[8], Wen Ye [9], Tiantian Yu[1], Weijian Liufu [1], Chao Cheng [4] ✉ & Yong Bao[1] ✉

[1]Department of Radiation Oncology, The First Affiliated Hospital of Sun Yat-Sen University, Guangzhou, China. [2]Department of Radiation Oncology, Shenzhen Qianhai Taikang Hospital, Shenzhen, China. [3]Department of Gastroenterology, The First Affiliated Hospital of Sun Yat-Sen University, Guangzhou, China. [4]Department of Thoracic Surgery, The First Affiliated Hospital of Sun Yat-Sen University, Guangzhou, China. [5]Gastrointestinal Surgery Center, The First Affiliated Hospital of Sun Yat-Sen University, Guangzhou, China. [6]Department of Radiology, The First Affiliated Hospital of Sun Yat-Sen University, Guangzhou, China. [7]Department of Nuclear Medicine, The First Affiliated Hospital of Sun Yat-Sen University, Guangzhou, China. [8]Department of Pathology, The First Affiliated Hospital of Sun Yat-Sen University, Guangzhou, China. [9]Department of Oncology, The First Affiliated Hospital of Sun Yat-Sen University, Guangzhou, China. [10]These authors contributed equally: Fang Peng, Jialiang Wu ✉e-mail: chengch3@mail.sysu.edu.cn; baoyong@mail.sysu.edu.cn

