## [Transparent Peer Review file · Nature Communications]

Induction chemotherapy plus camrelizumab followed by concurrent chemoradiotherapy in unresectable locally advanced esophageal squamous cell carcinoma: a single-arm phase II trial

Corresponding Author: Professor Yong Bao

Version 0:

Reviewer comments:

Reviewer #1

(Remarks to the Author)

This study investigates the efficacy and safety of induction chemotherapy (albumin-bound paclitaxel + carboplatin) combined with camrelizumab (anti-PD-1) followed by concurrent chemoradiotherapy (CCRT) in patients with unresectable locally advanced esophageal squamous cell carcinoma (ESCC). The data and analysis are comprehensive. The following problems should be resolved to make the conclusion more statistically defensible.

1. The sample size calculation ($n=49$) assumes a 14% improvement in 1-year OS (66% to 80%) with 80% power and one-sided $\alpha=0.05$. However, the rationale for selecting a one-sided test in this single-arm design is unclear. Phase 2 trials typically prioritize hypothesis generation, and a two-sided test might better align with exploratory objectives.
2. The exploratory biomarker analyses (PD-L1, CD8+, DR5+) and subgroup analyses (high/low responders, CR vs non-CR) do not appear to adjust for multiple comparisons. With numerous unplanned comparisons being made, the likelihood of type I error inflation is substantial. The authors should either apply appropriate multiplicity corrections or explicitly state these findings as hypothesis-generating.
3. Two deaths were attributed to COVID-19–related treatment interruptions. Sensitivity analyses excluding these patients could help assess the robustness of survival estimates.
4. While 95% CIs are provided for survival rates, key secondary endpoints (e.g., ORR, DCR) lack interval estimates. Report CIs for all efficacy outcomes would be better to quantify precision.

Reviewer #2

(Remarks to the Author)

This being a single arm study of 49 patients with locally advanced unresectable ESCC, makes it very difficult to figure out if there is any advantage. Judging objective response is also difficult but clinical response by PET and EGD is possible. Please provide this information. The follow up remains short. Any conclusion would be difficult to accept.

Page 7. no mention of PDL-1 expression or MS status

60GY of RT is not acceptable. 50.4 GY is standard.

Response data are difficult to understand. Does 89% partial response mean that all these patient were not disease-free?

Recommend a randomized trial

Please discuss KN975

Reviewer #3

(Remarks to the Author)

Overview

This phase 2, single-arm trial by Peng et al. evaluates the efficacy and safety of a sequential treatment regimen combining induction chemotherapy and camrelizumab followed by concurrent chemoradiotherapy (CCRT) in patients with unresectable, locally advanced esophageal squamous cell carcinoma (ESCC). This strategy seeks to improve on the historically poor outcomes associated with standard CCRT in this high-risk population.

Study Design

Conducted at the First Affiliated Hospital of Sun Yat-sen University, the trial enrolled 49 patients. Treatment included:

Two 21-day cycles of induction chemoimmunotherapy (albumin-bound paclitaxel, carboplatin, camrelizumab),

Followed by two 28-day cycles of fluorouracil and cisplatin with concurrent radiotherapy (50–63 Gy).

Primary endpoint: 1-year overall survival (OS) in the per-protocol set (PPS).

Secondary endpoints: Progression-free survival (PFS), overall response rate (ORR), disease control rate (DCR), safety, and quality of life.

Key Findings

ORR post-CCRT: 93.5%; DCR: 95.7%.

1-year OS: 87.0%; 2-year OS: 75.8% (PPS).

1-year and 2-year PFS: 71.7% and 62.9%, respectively.

Median OS and PFS were not reached at the time of analysis.

Tumor ulceration and necrosis improved significantly after induction therapy.

No cases of esophageal fistula; grade ≥ 3 esophagitis occurred in only 4.3% of patients.

Most common adverse event: Grade ≥ 3 lymphopenia (100%).

Immune and Biomarker Analysis

PD-L1 status was not predictive of response or survival.

High CD8+ T-cell infiltration and circulating CD3+CD8+ T-cell levels correlated with better OS and PFS.

Elevated DR5+ cell density in the tumor microenvironment also predicted improved outcomes, suggesting potential as a biomarker.

Strengths

Innovative integration of immunotherapy in the induction phase for unresectable ESCC.

Strong survival and response outcomes with manageable toxicity.

Inclusion of exploratory immunological and biomarker analyses.

Direct comparison to historical CRT controls highlights potential superiority.

Limitations

Single-arm design without a randomized comparator group.

Limited sample size and median follow-up; long-term durability of response yet to be determined.

Biomarker findings are exploratory and not validated.

COVID-19–related treatment interruptions may have influenced outcomes for a subset of patients.

Recommendation

While the study is well-conducted and offers promising data, I suggest the inclusion of a table summarizing the causes of death for the patients who died during the study. This would provide greater clarity on treatment-related versus disease-related mortality and help better contextualize the safety outcomes.

Conclusion

This study demonstrates that induction chemotherapy combined with camrelizumab followed by CCRT is a feasible and potentially superior alternative to standard CRT in unresectable locally advanced ESCC. With a high response rate, manageable adverse events, and favorable early survival outcomes, this regimen merits further validation in a randomized phase 3 trial. The exploratory findings regarding immune biomarkers also open the door to more personalized treatment strategies in the future.

Version 1:

Reviewer comments:

Reviewer #1

(Remarks to the Author)

The authors addressed my concerns, and the present version is more satisfactory.

Dear Reviewers:

Thank you for your insightful and constructive comments. We have addressed each of your questions, with detailed, point-by-point responses provided below. Corresponding modifications have been made to the manuscript and red-highlighted in the revised manuscript. Please review our responses and updated manuscript.

Best regards,

Yong Bao, Chao Cheng

Yong Bao

Department of Radiation Oncology

The First Affiliated Hospital of Sun Yat-sen University

Telephone: +86-20-87338692

E-mail: baoyong@mail.sysu.edu.cn

Chao Cheng

Department of Thoracic Surgery

The First Affiliated Hospital of Sun Yat-sen University

Telephone: +86-20-87338692

E-mail: chengch3@mail.sysu.edu.cn

Response to Reviewers:

Reviewer #1

Q1: The sample size calculation (n=49) assumes a 14% improvement in 1-year OS (66% to 80%) with 80% power and one-sided $\alpha=0.05$. However, the rationale for selecting a one-sided test in this single-arm design is unclear. Phase 2 trials typically prioritize hypothesis generation, and a two-sided test might better align with exploratory objectives.

Reply:

Thank you for your valuable comment regarding the sample size calculation.

We completely agree that a two-tailed test is generally more appropriate to calculate sample size when the efficacy advantage or disadvantage between two therapeutic regimens is uncertain. In the context of our study, the 1-year overall survival (OS) rate for standard chemoradiotherapy was based on the previous RTOG 94-05 study, which reported a rate of 66%. Based on our clinical practice, we observed that adding induction chemotherapy plus anti-PD-1 immunotherapy to standard chemoradiotherapy improved disease control and survival, a finding later confirmed by our retrospective study¹. Furthermore, it would be ethically implausible to assume an add-on regimen would perform worse than control. Therefore, we statistically assumed an increase in the 1-year OS rate in this study to 80%. Given this context, we chose to use a one-sided test for sample size calculation, a choice we believe is appropriate and reasonable for this trial.

Q2: The exploratory biomarker analyses (PD-L1, CD8+, DR5+) and subgroup analyses (high/low responders, CR vs non-CR) do not appear to adjust for multiple comparisons. With numerous unplanned comparisons being made, the likelihood of type I error inflation is substantial. The authors should either apply appropriate multiplicity corrections or explicitly state these findings as hypothesis-generating.

Reply:

Thank you for your valuable comment regarding multiplicity adjustments. As there is no formal hypothesis set for exploratory biomarker analyses, the p values here are all nominal, showing a trend for further demonstration. No type I error inflation here as no formal statistical decision could be made; multiplicity adjustment is not applicable. We have emphasized the necessity of further research for validation in the Discussion section (lines 359-360 on page 17 of the revised manuscript).

Q3: Two deaths were attributed to COVID-19–related treatment interruptions. Sensitivity analyses excluding these patients could help assess the robustness of survival estimates.

Reply:

Thank you for your valuable comment regarding sensitivity analyses.

We conducted a sensitivity analysis comparing overall survival (OS) and progression-free survival (PFS) between the original cohort and the sensitivity cohort (excluding two patients who experienced radiotherapy interruptions due to COVID-19) using Kaplan-Meier survival curves and log-rank tests. Results demonstrated that did not reverse the trend and the nominal p value did not indicate there is a difference clinically in OS (nominal $p = 0.914$) and PFS (nominal $p = 0.782$) between the two groups, indicating the robustness of survival estimates despite their inclusion in the primary analysis. These results have been added to the Results section and Supplementary Materials. We hope these revisions address your concerns. Thank you again for your constructive feedback. The specific modification can be found in the lines 137-140 on page 8 of the revised manuscript and Supplementary Figure S3.

Q4: While 95% CIs are provided for survival rates, key secondary endpoints (e.g., ORR, DCR) lack interval estimates. Report CIs for all efficacy outcomes would be better to quantify precision.

Reply:

Thank you for your valuable comment.

We fully agree with the necessity of including confidence intervals (CIs) for the key secondary efficacy endpoints. We have added confidence intervals (CIs) for these endpoints and the statistical method in the revised manuscript. Additionally, we have summarized the updated results in the response letter for your reference (Table 1). We hope these revisions address your concerns. Thank you again for your constructive feedback. The specific modification content can be found in the line 99 on page 6, lines 110-111 on page 6, lines 128-130 on page 7, lines 451-453 on page 21 of the revised manuscript and Table 1 of the Response Letter.

Reviewer #2

Q1: This being a single arm study of 49 patients with locally advanced unresectable ESCC, makes it very difficult to figure out if there is any advantage. Judging objective response is also difficult but clinical response by PET and EGD is possible. Please provide this information. The follow up remains short. Any conclusion would be difficult to accept.

Reply:

Thank you for your valuable comment.

We acknowledge that the single-arm phase II trial design, without a control arm, limits the strength of the findings, despite an exploratory post hoc analysis being conducted. We have explicitly emphasized the limitations of the results and the necessity for randomized controlled trials in the Discussion section (line 352 on page 16, lines 353-355 on page 17 of the revised manuscript).

We agree that clinical response assessed using PET and EGD would provide more robust evidence. However, due to the high cost of PET imaging and the lack of insurance coverage in China, PET imaging was not mandatory in this study. Instead, tumor response to induction therapy was assessed primarily by using enhanced CT of neck, chest, and upper abdomen, esophageal MRI, and esophagogastrosocopy with endoscopic ultrasound; while tumor response after CRT was assessed primarily by CT and MRI. The response of the primary esophageal tumor was evaluated based on the vertical length and maximal transverse thickness of the tumor, as defined by Conroy et al². When evaluating the overall response, the absence of PET-CT somewhat limited our ability to confirm CR in esophageal lesions. We have supplemented the examination measures for tumor response assessment in the Methods section and emphasized the limitations of evaluating CR in the Discussion section (lines 355-356 on page 17, lines 414-422 on page 18 of the revised manuscript).

As for the follow-up duration, the primary endpoint of our study was 1-year overall survival, and all patients have been followed up for longer (range: 15.8–42.2 months) than this predefined 1-year period. Of course, the short follow-up period is undeniable, as the median survival has not yet been reached. This has been addressed as a limitation in the Discussion section (lines 356-357 on page 17 of the revised manuscript).

Q2: Page 7. no mention of PD-L1 expression or MS status.

Reply:

We sincerely thank the reviewer for highlighting this important point.

Regarding PD-L1 expression, detailed analyses are presented in the “Tumor PD-L1 expression” subsection of the Results (line 208 on page 10, lines 209-216 on page 11 of the revised manuscript), where PD-L1 CPS scores and their association with clinical outcomes were evaluated. Although the data were incomplete, our exploratory analysis indicate no significant correlation between baseline PD-L1 expression and clinical outcomes such as CR rate, OS and PFS.

Concerning MS status, we acknowledge its potential relevance in efficacy of immunotherapy. According to the 2025 NCCN guidelines, MS testing is recommended for patients with unresectable locally advanced esophageal cancer. However, at the time of our study design, the NCCN guidelines suggested that these tests should be performed primarily when metastatic disease was suspected. Consequently, MS testing was not mandatory in our study protocol. Moreover, MSI is rare in esophageal squamous cell carcinoma, and its predictive role in immunotherapy for ESCC remains uncertain³⁻⁵; thus, it was not routinely assessed in this study. Future studies will include MS status testing to explore its prognostic value in this population.

Q3: 60GY of RT is not acceptable. 50.4 GY is standard.

Reply:

Thank you for your comment regarding the radiotherapy dose.

At study initiation, the 2019 NCCN Guidelines (Version 4) for Esophageal and Esophagogastric Junction Cancer recommended the definitive radiotherapy (RT) doses of 50-50.4 Gy (1.8-2.0 Gy per fraction). The NCCN guidelines also noted that higher doses (60-66 Gy) may be appropriate for cervical esophagus tumors⁶, particularly when surgery is not an option. No randomized evidence has definitively supported any benefit or detriment of higher doses compared to 50-50.4 Gy, which therefore remains the standard for CCRT in most Western countries.

In China, where esophageal squamous cell carcinoma (ESCC) accounts for 95% of all esophageal carcinoma cases⁷, clinicians frequently adopt 60 Gy dose delivered with modern radiotherapy technologies, considering that 50 Gy is inadequate for ESCC due to the different biological characteristics between ESCC and adenocarcinoma⁸. Concurrently in China, both the 2019 edition of the Chinese Society of Clinical Oncology (CSCO) Guidelines for the Diagnosis and Treatment of Esophageal Cancer and the 2019 Edition of the Chinese Guidelines for Radiotherapy of Esophageal Cancer recommend definitive concurrent chemoradiotherapy with a dose of 50-60 Gy using conventional fractionation. The primary rationale includes findings from prospective studies indicating no statistically significant difference in local control rates or survival rates between low-dose and high-dose radical radiotherapy groups. Furthermore, some retrospective

studies have shown that high-dose radiotherapy may improve the local control rate and survival rate for esophageal squamous cell carcinoma, although this remains controversial⁸⁻¹⁰.

Prospective evidence has evolved over time. The RTOG 85-01 trial established 50.4 Gy as the standard dose for definitive chemoradiotherapy in esophageal cancer. Subsequent trials (RTOG 94-05 trial¹¹, ARTDECO study¹² and a phase III trial by Ming Chen et al⁷) did not show a survival or local-control advantage with dose escalation. More recently, nevertheless, a newly published phase III trial reported a progression-free survival benefits in the high-dose group¹³, and several meta-analyses suggest that doses ≥ 60 Gy may improve local control rates and survival outcomes with manageable toxicities^{10,14,15}. Additionally, ongoing phase III trials are exploring varying dose strategies, such as KEYNOTE-975 (50 Gy or 60 Gy), KUNLUN (50–64 Gy), and RATIONALE-311 (50.4 Gy)^{16,17}. These studies underscore that the ideal dose in the IMRT era remains under active investigation.

Taken together, considering evolving evidence, current national guidelines endorsing 50–60 Gy, and widespread clinical practice in China, we believe that the use of 60 Gy in our trial is acceptable. We hope this explanation addresses your concerns.

Q4: Response data are difficult to understand. Does 89% partial response mean that all these patients were not disease-free?

Reply:

Thank you for your valuable comment regarding the tumor response.

According to the RECIST 1.1, partial response (PR) denotes $\geq 30\%$ decrease in the sum of diameters of target lesions, not complete resolution, whereas complete response (CR) requires the disappearance of all target lesions and any pathological lymph nodes (whether target or non-target) must have reduction in short axis to < 10 mm. Therefore, by definition, patients categorized as having a PR are not disease-free.

In our study, overall response after concurrent chemoradiotherapy was assessed primarily using contrast-enhanced CT of the neck, chest, and upper abdomen, combined with esophageal MRI. Tumor response of the primary esophageal lesion was based on both vertical length and maximal transverse thickness. We acknowledge that the absence of routine PET may have limited the detection of CR in some cases. This limitation has been discussed in the revised manuscript (lines 355-356 on page 17 of the revised manuscript).

Q5: Recommend a randomized trial.

Reply:

Thank you for your valuable comment regarding the suggestion to conduct a randomized trial.

We acknowledge that the single-arm phase II trial design, without a control arm, limits the strength of the findings, despite an exploratory post hoc analysis being conducted. A randomized Phase III trial is necessary to confirm these findings. We have explicitly emphasized the limitations of the results and the necessity of randomized controlled trials in the Discussion section. The specific modification can be found in the line 352 on page 16, lines 353-355 on page 17 of the revised manuscript.

Q6: Please discuss KN975.

Reply:

Thank you for your suggestion. A brief discussion of the KEYNOTE-975 trial has been incorporated into the revised Discussion section. The specific modification can be found in lines 300-304 on page 14, lines 305-313 on page 15 of the revised manuscript.

Reviewer #3

Q1: While the study is well-conducted and offers promising data, I suggest the inclusion of a table summarizing the causes of death for the patients who died during the study. This would provide greater clarity on treatment-related versus disease-related mortality and help better contextualize the safety outcomes.

Reply:

Thank you for your valuable comment regarding the inclusion of a table summarizing the causes of death. We fully agree with your suggestion, so we have included a table in the supplementary materials summarizing the causes of death for the patients. We believe this will help better contextualize the safety outcomes. The specific modification can be found in the Table S3 of supplementary materials.

Table 1. Key secondary efficacy endpoints with 95% confidence intervals

Endpoint	Rate	95% CI	Statistical Method
ORR (After induction chemotherapy plus camrelizumab)	91.3%	79.2%–97.6%	Clopper-Pearson
DCR (After induction chemotherapy plus camrelizumab)	100%	92.3%–100%	Clopper-Pearson
ORR (After one month of completing CCRT)	93.5%	82.1%–98.6%	Clopper-Pearson
DCR (After one month of completing CCRT)	95.7%	85.2%–99.5%	Clopper-Pearson
Tumor recurrence rate	26.5%	14.9%–41.1%	Clopper-Pearson
Locoregional failure rate	10.2%	3.4%–22.2%	Clopper-Pearson
Distant metastasis rate	8.2%	2.3%–19.6%	Clopper-Pearson
Simultaneous locoregional and distant failure rate	8.2%	2.3%–19.6%	Clopper-Pearson

CI confidence interval, *ORR* overall response rate, *DCR* disease control rate.

Reference

1. Lian, H. M. et al. Induction immunotherapy plus chemotherapy followed by definitive chemoradiation therapy in locally advanced esophageal squamous cell carcinoma: a propensity-score matched study. *Cancer Immunol. Immunother.* **73**, 55 (2024).
2. Conroy, T. et al. Definitive chemoradiotherapy with FOLFOX versus fluorouracil and cisplatin in patients with oesophageal cancer (PRODIGE5/ACCORD17): final results of a randomised, phase 2/3 trial. *Lancet Oncol.* **15**, 305-314 (2014).
3. Salem, M. E. et al. Comparative Molecular Analyses of Esophageal Squamous Cell Carcinoma, Esophageal Adenocarcinoma, and Gastric Adenocarcinoma. *Oncologist* **23**, 1319-1327 (2018).
4. Matsumoto, Y. et al. Microsatellite instability and clinicopathological features in esophageal squamous cell cancer. *Oncol. Rep.* **18**, 1123-1127 (2007).
5. Hayashi, M. et al. Microsatellite instability in esophageal squamous cell carcinoma is not associated with hMLH1 promoter hypermethylation. *Pathol. Int.* **53**, 270-276 (2003).
6. Wang, S. et al. Esophageal cancer located at the neck and upper thorax treated with concurrent chemoradiation: a single-institution experience. *J. Thorac. Oncol.* **1**, 252-259 (2006).
7. Xu, Y. et al. A Phase III Multicenter Randomized Clinical Trial of 60 Gy versus 50 Gy Radiation Dose in Concurrent Chemoradiotherapy for Inoperable Esophageal Squamous Cell Carcinoma. *Clin. Cancer Res.* **28**, 1792-1799 (2022).
8. Zhang, W. et al. Dose-escalated radiotherapy improved survival for esophageal cancer patients with a clinical complete response after standard-dose radiotherapy with concurrent chemotherapy. *Cancer Manag. Res.* **10**, 2675-2682 (2018).
9. Kim, H. J. et al. Dose-Response Relationship between Radiation Dose and Loco-regional Control in Patients with Stage II-III Esophageal Cancer Treated with Definitive Chemoradiotherapy. *Cancer Res. Treat.* **49**, 669-677 (2017).
10. Song, T., Liang, X., Fang, M. & Wu, S. High-dose versus conventional-dose irradiation in cisplatin-based definitive concurrent chemoradiotherapy for esophageal cancer: a systematic review and pooled analysis. *Expert Rev. Anticancer Ther.* **15**, 1157-1169 (2015).
11. Minsky, B. D. et al. INT 0123 (Radiation Therapy Oncology Group 94-05) phase III trial of combined-modality therapy for esophageal cancer: high-dose versus standard-dose radiation therapy. *J. Clin. Oncol.* **20**, 1167-1174 (2002).
12. Hulshof, M. et al. Randomized Study on Dose Escalation in Definitive Chemoradiation for Patients With Locally Advanced Esophageal Cancer (ARTDECO Study). *J. Clin. Oncol.* **39**, 2816-2824 (2021).
13. Zhang, J. et al. Concurrent chemoradiotherapy of different radiation doses and different irradiation fields for locally advanced thoracic esophageal squamous cell carcinoma: A randomized, multicenter, phase III clinical trial. *Cancer Commun (Lond)* **44**, 1173-1188 (2024).

14. Chang, C. L. et al. Dose escalation intensity-modulated radiotherapy-based concurrent chemoradiotherapy is effective for advanced-stage thoracic esophageal squamous cell carcinoma. *Radiother. Oncol.* **125**, 73-79 (2017).
15. Chow, R., Lock, M., Lee, S. L., Lo, S. S. & Simone, C. B., 2nd. Esophageal Cancer Radiotherapy Dose Escalation Meta Regression Commentary: "High vs. Low Radiation Dose of Concurrent Chemoradiotherapy for Esophageal Carcinoma With Modern Radiotherapy Techniques: A Meta-Analysis". *Front. Oncol.* **11**, 700300 (2021).
16. Shah, M. A. et al. KEYNOTE-975 study design: a Phase III study of definitive chemoradiotherapy plus pembrolizumab in patients with esophageal carcinoma. *Future Oncol.* **17**, 1143-1153 (2021).
17. Wang, L. et al. A phase 3 randomized, double-blind, placebo-controlled, multicenter, global study of durvalumab with and after chemoradiotherapy in patients with locally advanced, unresectable esophageal squamous cell carcinoma: KUNLUN. *J. Clin. Oncol.* **40**, TPS373 (2022).

Dear Reviewers:

Thank you for your insightful and constructive comments. We have addressed each of your questions, with detailed, point-by-point responses provided below. Corresponding modifications have been made to the manuscript and red-highlighted in the revised manuscript. Please review our responses and updated manuscript.

Best regards,

Yong Bao, Chao Cheng

Yong Bao

Department of Radiation Oncology

The First Affiliated Hospital of Sun Yat-sen University

Telephone: +86-20-87338692

E-mail: baoyong@mail.sysu.edu.cn

Chao Cheng

Department of Thoracic Surgery

The First Affiliated Hospital of Sun Yat-sen University

Telephone: +86-20-87338692

E-mail: chengch3@mail.sysu.edu.cn

Response to Reviewers:

Reviewer #1

Q1: The sample size calculation (n=49) assumes a 14% improvement in 1-year OS (66% to 80%) with 80% power and one-sided $\alpha=0.05$. However, the rationale for selecting a one-sided test in this single-arm design is unclear. Phase 2 trials typically prioritize hypothesis generation, and a two-sided test might better align with exploratory objectives.

Reply:

Thank you for your valuable comment regarding the sample size calculation.

We completely agree that a two-tailed test is generally more appropriate to calculate sample size when the efficacy advantage or disadvantage between two therapeutic regimens is uncertain. In the context of our study, the 1-year overall survival (OS) rate for standard chemoradiotherapy was based on the previous RTOG 94-05 study, which reported a rate of 66%. Based on our clinical practice, we observed that adding induction chemotherapy plus anti-PD-1 immunotherapy to standard chemoradiotherapy improved disease control and survival, a finding later confirmed by our retrospective study¹. Furthermore, it would be ethically implausible to assume an add-on regimen would perform worse than control. Therefore, we statistically assumed an increase in the 1-year OS rate in this study to 80%. Given this context, we chose to use a one-sided test for sample size calculation, a choice we believe is appropriate and reasonable for this trial.

Q2: The exploratory biomarker analyses (PD-L1, CD8+, DR5+) and subgroup analyses (high/low responders, CR vs non-CR) do not appear to adjust for multiple comparisons. With numerous unplanned comparisons being made, the likelihood of type I error inflation is substantial. The authors should either apply appropriate multiplicity corrections or explicitly state these findings as hypothesis-generating.

Reply:

Thank you for your valuable comment regarding multiplicity adjustments. As there is no formal hypothesis set for exploratory biomarker analyses, the p values here are all nominal, showing a trend for further demonstration. No type I error inflation here as no formal statistical decision could be made; multiplicity adjustment is not applicable. We have emphasized the necessity of further research for validation in the Discussion section (lines 359-360 on page 17 of the revised manuscript).

Q3: Two deaths were attributed to COVID-19–related treatment interruptions. Sensitivity analyses excluding these patients could help assess the robustness of survival estimates.

Reply:

Thank you for your valuable comment regarding sensitivity analyses.

We conducted a sensitivity analysis comparing overall survival (OS) and progression-free survival (PFS) between the original cohort and the sensitivity cohort (excluding two patients who experienced radiotherapy interruptions due to COVID-19) using Kaplan-Meier survival curves and log-rank tests. Results demonstrated that did not reverse the trend and the nominal p value did not indicate there is a difference clinically in OS (nominal $p = 0.914$) and PFS (nominal $p = 0.782$) between the two groups, indicating the robustness of survival estimates despite their inclusion in the primary analysis. These results have been added to the Results section and Supplementary Materials. We hope these revisions address your concerns. Thank you again for your constructive feedback. The specific modification can be found in the lines 137-140 on page 8 of the revised manuscript and Supplementary Figure S3.

Q4: While 95% CIs are provided for survival rates, key secondary endpoints (e.g., ORR, DCR) lack interval estimates. Report CIs for all efficacy outcomes would be better to quantify precision.

Reply:

Thank you for your valuable comment.

We fully agree with the necessity of including confidence intervals (CIs) for the key secondary efficacy endpoints. We have added confidence intervals (CIs) for these endpoints and the statistical method in the revised manuscript. Additionally, we have summarized the updated results in the response letter for your reference (Table 1). We hope these revisions address your concerns. Thank you again for your constructive feedback. The specific modification content can be found in the line 99 on page 6, lines 110-111 on page 6, lines 128-130 on page 7, lines 451-453 on page 21 of the revised manuscript and Table 1 of the Response Letter.

Reviewer #2

Q1: This being a single arm study of 49 patients with locally advanced unresectable ESCC, makes it very difficult to figure out if there is any advantage. Judging objective response is also difficult but clinical response by PET and EGD is possible. Please provide this information. The follow up remains short. Any conclusion would be difficult to accept.

Reply:

Thank you for your valuable comment.

We acknowledge that the single-arm phase II trial design, without a control arm, limits the strength of the findings, despite an exploratory post hoc analysis being conducted. We have explicitly emphasized the limitations of the results and the necessity for randomized controlled trials in the Discussion section (line 352 on page 16, lines 353-355 on page 17 of the revised manuscript).

We agree that clinical response assessed using PET and EGD would provide more robust evidence. However, due to the high cost of PET imaging and the lack of insurance coverage in China, PET imaging was not mandatory in this study. Instead, tumor response to induction therapy was assessed primarily by using enhanced CT of neck, chest, and upper abdomen, esophageal MRI, and esophagogastrosocopy with endoscopic ultrasound; while tumor response after CRT was assessed primarily by CT and MRI. The response of the primary esophageal tumor was evaluated based on the vertical length and maximal transverse thickness of the tumor, as defined by Conroy et al². When evaluating the overall response, the absence of PET-CT somewhat limited our ability to confirm CR in esophageal lesions. We have supplemented the examination measures for tumor response assessment in the Methods section and emphasized the limitations of evaluating CR in the Discussion section (lines 355-356 on page 17, lines 414-422 on page 18 of the revised manuscript).

As for the follow-up duration, the primary endpoint of our study was 1-year overall survival, and all patients have been followed up for longer (range: 15.8–42.2 months) than this predefined 1-year period. Of course, the short follow-up period is undeniable, as the median survival has not yet been reached. This has been addressed as a limitation in the Discussion section (lines 356-357 on page 17 of the revised manuscript).

Q2: Page 7. no mention of PD-L1 expression or MS status.

Reply:

We sincerely thank the reviewer for highlighting this important point.

Regarding PD-L1 expression, detailed analyses are presented in the “Tumor PD-L1 expression” subsection of the Results (line 208 on page 10, lines 209-216 on page 11 of the revised manuscript), where PD-L1 CPS scores and their association with clinical outcomes were evaluated. Although the data were incomplete, our exploratory analysis indicate no significant correlation between baseline PD-L1 expression and clinical outcomes such as CR rate, OS and PFS.

Concerning MS status, we acknowledge its potential relevance in efficacy of immunotherapy. According to the 2025 NCCN guidelines, MS testing is recommended for patients with unresectable locally advanced esophageal cancer. However, at the time of our study design, the NCCN guidelines suggested that these tests should be performed primarily when metastatic disease was suspected. Consequently, MS testing was not mandatory in our study protocol. Moreover, MSI is rare in esophageal squamous cell carcinoma, and its predictive role in immunotherapy for ESCC remains uncertain³⁻⁵; thus, it was not routinely assessed in this study. Future studies will include MS status testing to explore its prognostic value in this population.

Q3: 60GY of RT is not acceptable. 50.4 GY is standard.

Reply:

Thank you for your comment regarding the radiotherapy dose.

At study initiation, the 2019 NCCN Guidelines (Version 4) for Esophageal and Esophagogastric Junction Cancer recommended the definitive radiotherapy (RT) doses of 50-50.4 Gy (1.8-2.0 Gy per fraction). The NCCN guidelines also noted that higher doses (60-66 Gy) may be appropriate for cervical esophagus tumors⁶, particularly when surgery is not an option. No randomized evidence has definitively supported any benefit or detriment of higher doses compared to 50-50.4 Gy, which therefore remains the standard for CCRT in most Western countries.

In China, where esophageal squamous cell carcinoma (ESCC) accounts for 95% of all esophageal carcinoma cases⁷, clinicians frequently adopt 60 Gy dose delivered with modern radiotherapy technologies, considering that 50 Gy is inadequate for ESCC due to the different biological characteristics between ESCC and adenocarcinoma⁸. Concurrently in China, both the 2019 edition of the Chinese Society of Clinical Oncology (CSCO) Guidelines for the Diagnosis and Treatment of Esophageal Cancer and the 2019 Edition of the Chinese Guidelines for Radiotherapy of Esophageal Cancer recommend definitive concurrent chemoradiotherapy with a dose of 50-60 Gy using conventional fractionation. The primary rationale includes findings from prospective studies indicating no statistically significant difference in local control rates or survival rates between low-dose and high-dose radical radiotherapy groups. Furthermore, some retrospective

studies have shown that high-dose radiotherapy may improve the local control rate and survival rate for esophageal squamous cell carcinoma, although this remains controversial⁸⁻¹⁰.

Prospective evidence has evolved over time. The RTOG 85-01 trial established 50.4 Gy as the standard dose for definitive chemoradiotherapy in esophageal cancer. Subsequent trials (RTOG 94-05 trial¹¹, ARTDECO study¹² and a phase III trial by Ming Chen et al⁷) did not show a survival or local-control advantage with dose escalation. More recently, nevertheless, a newly published phase III trial reported a progression-free survival benefits in the high-dose group¹³, and several meta-analyses suggest that doses ≥ 60 Gy may improve local control rates and survival outcomes with manageable toxicities^{10,14,15}. Additionally, ongoing phase III trials are exploring varying dose strategies, such as KEYNOTE-975 (50 Gy or 60 Gy), KUNLUN (50–64 Gy), and RATIONALE-311 (50.4 Gy)^{16,17}. These studies underscore that the ideal dose in the IMRT era remains under active investigation.

Taken together, considering evolving evidence, current national guidelines endorsing 50–60 Gy, and widespread clinical practice in China, we believe that the use of 60 Gy in our trial is acceptable. We hope this explanation addresses your concerns.

Q4: Response data are difficult to understand. Does 89% partial response mean that all these patients were not disease-free?

Reply:

Thank you for your valuable comment regarding the tumor response.

According to the RECIST 1.1, partial response (PR) denotes $\geq 30\%$ decrease in the sum of diameters of target lesions, not complete resolution, whereas complete response (CR) requires the disappearance of all target lesions and any pathological lymph nodes (whether target or non-target) must have reduction in short axis to < 10 mm. Therefore, by definition, patients categorized as having a PR are not disease-free.

In our study, overall response after concurrent chemoradiotherapy was assessed primarily using contrast-enhanced CT of the neck, chest, and upper abdomen, combined with esophageal MRI. Tumor response of the primary esophageal lesion was based on both vertical length and maximal transverse thickness. We acknowledge that the absence of routine PET may have limited the detection of CR in some cases. This limitation has been discussed in the revised manuscript (lines 355-356 on page 17 of the revised manuscript).

Q5: Recommend a randomized trial.

Reply:

Thank you for your valuable comment regarding the suggestion to conduct a randomized trial.

We acknowledge that the single-arm phase II trial design, without a control arm, limits the strength of the findings, despite an exploratory post hoc analysis being conducted. A randomized Phase III trial is necessary to confirm these findings. We have explicitly emphasized the limitations of the results and the necessity of randomized controlled trials in the Discussion section. The specific modification can be found in the line 352 on page 16, lines 353-355 on page 17 of the revised manuscript.

Q6: Please discuss KN975.

Reply:

Thank you for your suggestion. A brief discussion of the KEYNOTE-975 trial has been incorporated into the revised Discussion section. The specific modification can be found in lines 300-304 on page 14, lines 305-313 on page 15 of the revised manuscript.

Reviewer #3

Q1: While the study is well-conducted and offers promising data, I suggest the inclusion of a table summarizing the causes of death for the patients who died during the study. This would provide greater clarity on treatment-related versus disease-related mortality and help better contextualize the safety outcomes.

Reply:

Thank you for your valuable comment regarding the inclusion of a table summarizing the causes of death. We fully agree with your suggestion, so we have included a table in the supplementary materials summarizing the causes of death for the patients. We believe this will help better contextualize the safety outcomes. The specific modification can be found in the Table S3 of supplementary materials.

Table 1. Key secondary efficacy endpoints with 95% confidence intervals

Endpoint	Rate	95% CI	Statistical Method
ORR (After induction chemotherapy plus camrelizumab)	91.3%	79.2%–97.6%	Clopper-Pearson
DCR (After induction chemotherapy plus camrelizumab)	100%	92.3%–100%	Clopper-Pearson
ORR (After one month of completing CCRT)	93.5%	82.1%–98.6%	Clopper-Pearson
DCR (After one month of completing CCRT)	95.7%	85.2%–99.5%	Clopper-Pearson
Tumor recurrence rate	26.5%	14.9%–41.1%	Clopper-Pearson
Locoregional failure rate	10.2%	3.4%–22.2%	Clopper-Pearson
Distant metastasis rate	8.2%	2.3%–19.6%	Clopper-Pearson
Simultaneous locoregional and distant failure rate	8.2%	2.3%–19.6%	Clopper-Pearson

CI confidence interval, *ORR* overall response rate, *DCR* disease control rate.

Reference

1. Lian, H. M. et al. Induction immunotherapy plus chemotherapy followed by definitive chemoradiation therapy in locally advanced esophageal squamous cell carcinoma: a propensity-score matched study. *Cancer Immunol. Immunother.* **73**, 55 (2024).
2. Conroy, T. et al. Definitive chemoradiotherapy with FOLFOX versus fluorouracil and cisplatin in patients with oesophageal cancer (PRODIGE5/ACCORD17): final results of a randomised, phase 2/3 trial. *Lancet Oncol.* **15**, 305-314 (2014).
3. Salem, M. E. et al. Comparative Molecular Analyses of Esophageal Squamous Cell Carcinoma, Esophageal Adenocarcinoma, and Gastric Adenocarcinoma. *Oncologist* **23**, 1319-1327 (2018).
4. Matsumoto, Y. et al. Microsatellite instability and clinicopathological features in esophageal squamous cell cancer. *Oncol. Rep.* **18**, 1123-1127 (2007).
5. Hayashi, M. et al. Microsatellite instability in esophageal squamous cell carcinoma is not associated with hMLH1 promoter hypermethylation. *Pathol. Int.* **53**, 270-276 (2003).
6. Wang, S. et al. Esophageal cancer located at the neck and upper thorax treated with concurrent chemoradiation: a single-institution experience. *J. Thorac. Oncol.* **1**, 252-259 (2006).
7. Xu, Y. et al. A Phase III Multicenter Randomized Clinical Trial of 60 Gy versus 50 Gy Radiation Dose in Concurrent Chemoradiotherapy for Inoperable Esophageal Squamous Cell Carcinoma. *Clin. Cancer Res.* **28**, 1792-1799 (2022).
8. Zhang, W. et al. Dose-escalated radiotherapy improved survival for esophageal cancer patients with a clinical complete response after standard-dose radiotherapy with concurrent chemotherapy. *Cancer Manag. Res.* **10**, 2675-2682 (2018).
9. Kim, H. J. et al. Dose-Response Relationship between Radiation Dose and Loco-regional Control in Patients with Stage II-III Esophageal Cancer Treated with Definitive Chemoradiotherapy. *Cancer Res. Treat.* **49**, 669-677 (2017).
10. Song, T., Liang, X., Fang, M. & Wu, S. High-dose versus conventional-dose irradiation in cisplatin-based definitive concurrent chemoradiotherapy for esophageal cancer: a systematic review and pooled analysis. *Expert Rev. Anticancer Ther.* **15**, 1157-1169 (2015).
11. Minsky, B. D. et al. INT 0123 (Radiation Therapy Oncology Group 94-05) phase III trial of combined-modality therapy for esophageal cancer: high-dose versus standard-dose radiation therapy. *J. Clin. Oncol.* **20**, 1167-1174 (2002).
12. Hulshof, M. et al. Randomized Study on Dose Escalation in Definitive Chemoradiation for Patients With Locally Advanced Esophageal Cancer (ARTDECO Study). *J. Clin. Oncol.* **39**, 2816-2824 (2021).
13. Zhang, J. et al. Concurrent chemoradiotherapy of different radiation doses and different irradiation fields for locally advanced thoracic esophageal squamous cell carcinoma: A randomized, multicenter, phase III clinical trial. *Cancer Commun (Lond)* **44**, 1173-1188 (2024).

14. Chang, C. L. et al. Dose escalation intensity-modulated radiotherapy-based concurrent chemoradiotherapy is effective for advanced-stage thoracic esophageal squamous cell carcinoma. *Radiother. Oncol.* **125**, 73-79 (2017).
15. Chow, R., Lock, M., Lee, S. L., Lo, S. S. & Simone, C. B., 2nd. Esophageal Cancer Radiotherapy Dose Escalation Meta Regression Commentary: "High vs. Low Radiation Dose of Concurrent Chemoradiotherapy for Esophageal Carcinoma With Modern Radiotherapy Techniques: A Meta-Analysis". *Front. Oncol.* **11**, 700300 (2021).
16. Shah, M. A. et al. KEYNOTE-975 study design: a Phase III study of definitive chemoradiotherapy plus pembrolizumab in patients with esophageal carcinoma. *Future Oncol.* **17**, 1143-1153 (2021).
17. Wang, L. et al. A phase 3 randomized, double-blind, placebo-controlled, multicenter, global study of durvalumab with and after chemoradiotherapy in patients with locally advanced, unresectable esophageal squamous cell carcinoma: KUNLUN. *J. Clin. Oncol.* **40**, TPS373 (2022).